# The integration of metabolic and proteomic data uncovers an augmentation of the sphingolipid biosynthesis pathway during T-cell differentiation
Toshio Kanno [1], Ryo Konno[2], Masaru Sato[3], Atsushi Kurabayashi[3], Keisuke Miyako[2], Takahiro Nakajima[1], Satoru Yokoyama[1], Shigemi Sasamoto[1], Hikari K. Asou[1], Junichiro Ohzeki[1], Yoshinori Hasegawa[2], Kazutaka Ikeda[2], Yusuke Kawashima [2], Osamu Ohara [2] & Yusuke Endo [1,4] ✉

Recent studies have highlighted the significance of cellular metabolism in the initiation of clonal expansion and effector differentiation of T cells. Upon exposure to antigens, naïve CD4+ T cells undergo metabolic reprogramming to meet their metabolic requirements. However, only few studies have simultaneously evaluated the changes in protein and metabolite levels during T cell differentiation. Our research seeks to fill the gap by conducting a comprehensive analysis of changes in levels of metabolites, including sugars, amino acids, intermediates of the TCA cycle, fatty acids, and lipids. By integrating metabolomics and proteomics data, we discovered that the quantity and composition of cellular lipids underwent significant changes in different effector Th cell subsets. Especially, we found that the sphingolipid biosynthesis pathway was commonly activated in Th1, Th2, Th17, and iTreg cells and that inhibition of this pathway led to the suppression of Th17 and iTreg cells differentiation. Additionally, we discovered that Th17 and iTreg cells enhance glycosphingolipid metabolism, and inhibition of this pathway also results in the suppression of Th17 and iTreg cell generation. These findings demonstrate that the utility of our combined metabolomics and proteomics analysis in furthering the understanding of metabolic transition during Th cell differentiation.

T cells play a central role in acquired immunity by recognizing foreign antigens through T-cell antigen receptor (TCR). After antigenic stimulation, quiescent naïve CD4+ T cells drives rapid proliferation and acquire an effector phenotype[1,2]. In particular, lineage-specifying cytokines determine the differentiation of naïve CD4+ T cells into functionally distinct subsets, including Th1, Th2, Th17, and regulatory T cells (Tregs)[1,2]. The differentiation of Th1 cells is induced by treatment with IL-12, leading to an increase in the production of IFNγ for anti-viral responses. Th2 cell differentiation, in the presence of IL-4, results in the production IL-4, IL-5, and IL-13, thereby regulating both parasite infections and allergic inflammation. The differentiation of Th17 cells requires stimulation with both IL-6 and TGFβ, which leads to the upregulation of IL-17A and IL-17F and confers protection against fungal pathogens. The differentiation of Treg cells, crucial for maintaining immunological tolerance and immune homeostasis, is dependent on TGFβ and IL-2 stimulation. Dysregulation of effector T cell response has been associated with the progression of cancer and autoimmune diseases[1–3].

In recent years, increasing evidence has highlighted the significance of cellular metabolism in many aspects of T-cell biology[1–3]. In naïve CD8+ T cells, the metabolic switch from oxidative phosphorylation (OXPHOS) to glycolysis is essential for the acquisition of effector function following antigenic stimulation[4]. It is also reported that cell intrinsic lysosomal lipolysis is necessary for the generation and maintenance of memory CD8+ T cells[5]. In CD4+ T cells, studies have demonstrated that the changes in intracellular metabolism are closely linked to the regulation of immune function[1,2,6,7]. Activated Th cells have been observed to induce heightened nutrient uptake and metabolic rates, resulting in a significant increase of glucose and amino acid transport[6,7]. The glucose transporter *Glut1*, has been

[1]Department of Frontier Research and Development, Laboratory of Medical Omics Research, Kazusa DNA Research Institute, Kisarazu, Chiba, Japan. [2]Department of Applied Genomics Kazusa DNA Research Institute, Kisarazu, Chiba, Japan. [3]Department of Research and Development, Kazusa DNA Research Institutes, Kisarazu, Japan. [4]Department of Omics Medicine, Graduate School of Medicine, Chiba University, Chiba, Japan. ✉e-mail: endo@kazusa.or.jp

found to play a critical role in T cell activation, as its deficiency has been shown to prevent increased glucose uptake and glycolysis, leading to the suppression of TCR stimulation-induced proliferation and the inhibition of Teff cell survival and differentiation[6,7]. Additionally, a deficiency in the glutamine transporter, ASCT2, has been found to negatively impact both Th1 and Th17 cell specification, while promoting the generation of Treg cells[8]. Furthermore, the restriction of serine metabolism and the inhibition of serine hydroxymethyltransferase 1 (SHMT1) caused a suppression of optimal cell proliferation[9].

Besides the regulation of glucose and amino acid metabolism, lipid metabolism has been identified as a key regulator in controlling T cell responses. The mTOR-PPARγ axis-mediated fatty acid metabolic reprogramming is required for the early activation of CD4[+] T cells[10]. The Acetyl-CoA Carboxylase 1 (ACC1) enzyme, which acts as a rate-limiting enzyme in fatty acid biosynthesis, serves as a marker of the memory potential of individual CD4[+] T cells. Pharmacological inhibition or genetic deletion of ACC1 have been shown to impair the differentiation of pathogenic Th2 and Th17 cells[11,12]. Middle- and long-chain fatty acids are crucial for Th1 and Th17 cell differentiation, whereas short-chain fatty acids promote Treg cell differentiation[13]. Recent study also highlights the dependence of Tregs on fatty acid oxidation (FAO) and OXPHOS for their homeostatic maintenance and suppressive capacity[14]. Despite numerous studies exploring the relationship between immune function and metabolism in CD4[+] T cells, limited research has been conducted to evaluate the metabolic changes occurring during T-cell differentiation, specifically at the protein and metabolites levels.

The growing application of various omics technologies led us to explore different layer of biological regulation, including transcriptome, proteome, and metabolome[15–19]. Previously, quantitative proteomic analyses were performed to explore the influence of TCR activation on protein ubiquitination and phosphorylation[15,16]. A combined analysis of protein expression and ubiquitination status was performed to establish a predictive framework to assess the relationship between ubiquitylation and protein abundance[15]. Furthermore, analyses of whole proteomes and phospho-proteomes have revealed early dynamic phosphorylation events following two hours of TCR stimulation and a subsequent amplification of both protein phosphorylation and expression[16]. Omics analyses have also been performed to determine the underlying regulatory mechanisms of T cell differentiation[17–19]. An integrative analysis of proteomics, bulk RNA-seq, and single-cell RNA-seq was performed to compare the differences in cytokine responses between human naïve and memory CD4[+] T cells[17]. Recently, Sen et al. conducted an integrative analysis of quantitative metabolome and published gene expression data to investigate metabolic changes specific to effector T cell subsets[18]. Although the multitude of omics analyses that have been conducted, only few studies have simultaneously evaluated the changes in protein and metabolite levels during T cell differentiation.

In this study, we have conducted deeper untargeted metabolomics analysis to investigate the fluctuating alterations in cellular metabolites during the differentiation of effector Th cells. We then integrated the metabolomic data with our previously published proteomic data, allowing us the opportunity to evaluate the metabolic shifts in both metabolite levels and synthase[19]. The results of the integrative analysis revealed that TCR stimulation significantly elevated the biosynthesis pathway of sphingolipids. Furthermore, Th17 and iTreg cells were found to augment glycosphingolipid metabolism and the inhibition of sphingolipid or glycosphingolipid biosynthesis prevented the differentiation of Th17 and iTreg cells. Our findings thus demonstrate that the biosynthesis of sphingolipids and glycosphingolipids regulates the generation of Th17 and iTreg cells.

## Results
### TCR stimulation caused dynamic changes in the intracellular metabolism, including glucose, amino acid, intermediates of TCA cycle, and fatty acid metabolism to fuel biosynthetic processes

To evaluate the changes in metabolic pathways during T cell differentiation, we first analyzed the amounts of intracellular metabolites and

expression of metabolic proteins using a mass-spectrum-based metabolomics approach and previously established proteogenomic database[19] (Fig. 1a). Naïve CD4[+] T cells were stimulated with immobilized the anti-TCR mAb and anti-CD28 mAb for 48 h under Th0, Th1, Th2, Th17, or induced regulatory T cell (iTregs) culture conditions. Dead cell removal was performed before sample preparation for omics analysis. As reported previously, our omics analysis showed that TCR stimulation elicited a rapid induction of glycolysis at the level of metabolites and proteins[6,7] (Fig. 1b). The intermediate of glycolysis, G3P, was decreased, while the end product, lactate, displayed a substantial increase upon TCR stimulation (Fig. 1b). Proteomics data revealed significant increases in the expression of glucose transporters and glycolytic enzymes, including GLUT1, GLUT3, HK1, GPI, PKM, and LDHA. Furthermore, we observed that TCR stimulation enhanced the utilization of glucogenic amino acids synthesis, which was synthesized at lower levels in resting naïve CD4[+] T cells (Fig. 1b lower panel, and Supplementary Fig. 1a). While the amounts of serine was reduced by 1.5-fold in Th0, Th1, and Th2 cells and increased by 1.5-fold in Th17 and iTreg cells, the amounts of glycine and cysteine was significantly elevated in each effector Th cell subsets (Fig. 1b, upper panel). In accordance with this result, protein expression of glucogenic amino acids synthase was highly upregulated. These amino acids serve as a primary source for the one carbon pathway, an essential component for the biosynthesis of both proteins and DNA. Thus, these findings suggest that activated Th cells acquire one carbon metabolism to support their rapid proliferation. Next, we focused on TCA cycle and its associated metabolites. Despite observed minimal variation in the intermediates of the TCA cycle, such as succinate, fumarate, malate, and citrate, between naïve CD4[+] T cells and effector Th cell subsets, the amounts of glutamine were greatly reduced following TCR stimulation (Fig. 1c). This finding was supported by the decrease in expression of the enzyme glutamine synthase (GLUL) and the concurrent increase of degradative enzyme (GLS2) in effector Th cell subsets, demonstrating the validity of our multi-omics analysis in reflecting the activation of glutaminolysis in effector Th cells[20]. In addition, the level of glutamate was increased in Th17 cells, whereas no significant change was observed in the amounts of glutamate in the other effector Th cell subsets. Furthermore, there was a significant increase in the levels of several amino acids, including alanine, lysine leucine, threonine, aspartate, asparagine, phenylalanine, tyrosine, and proline. Correspondingly, we also noted an up-regulation in the expression of their synthase and transporters, indicating that amino acid biosynthesis and uptake were activated during T cell activation (Fig. 1b, c and Supplementary Fig. 1b). It is noteworthy that we observed a marked increase in the amounts of β-alanine, a crucial component for the production of acetyl-CoA, in effector Th cell subsets (Fig. 1d). Since acetyl-CoA was required for the fatty acid biosynthesis, we next investigated the effect of TCR-stimulation on fatty acid biosynthesis pathway. Initially, we detected minor alterations in the expression of enzymes associated with fatty acid elongation, such as ELOVL1, ELOVL5, ELOVL6, and ELOVL7 in Th cell subsets (Fig. 1e). In contrast, TCR stimulation resulted in a marked upregulation of proteins involved in fatty acid desaturation, including SCD1, SCD2, FADS1, and FADS2 (Fig. 1e). Subsequently, we sought to assess the levels of fatty acid saturation, focusing on saturated fatty acid (SFA), mono-unsaturated fatty acid (MUFA), and polyunsaturated fatty acid (PUFA). In quiescent naïve CD4[+] T cells, SFA accounted for 92.5% of total fatty acids, while MUFAs and PUFAs comprised relatively smaller proportions (MUFA: 2.68%, PUFA: 4.85%) (Fig. 1f). Activated Th cell exhibited changes in the degree of fatty acid saturation, with the MUFA composition of activated Th cell subsets increasing to 26-37% (Fig. 1f). These findings are reflected in the observation that while most fatty acids were largely increased by TCR-stimulation, the amounts of MUFAs increased the most (Fig. 1g). Taken together, these data indicate that TCR stimulation confers activated Th cells with an increased ability to engage in glycolysis and amino acid synthesis and generates a diversity of fatty acid composition.

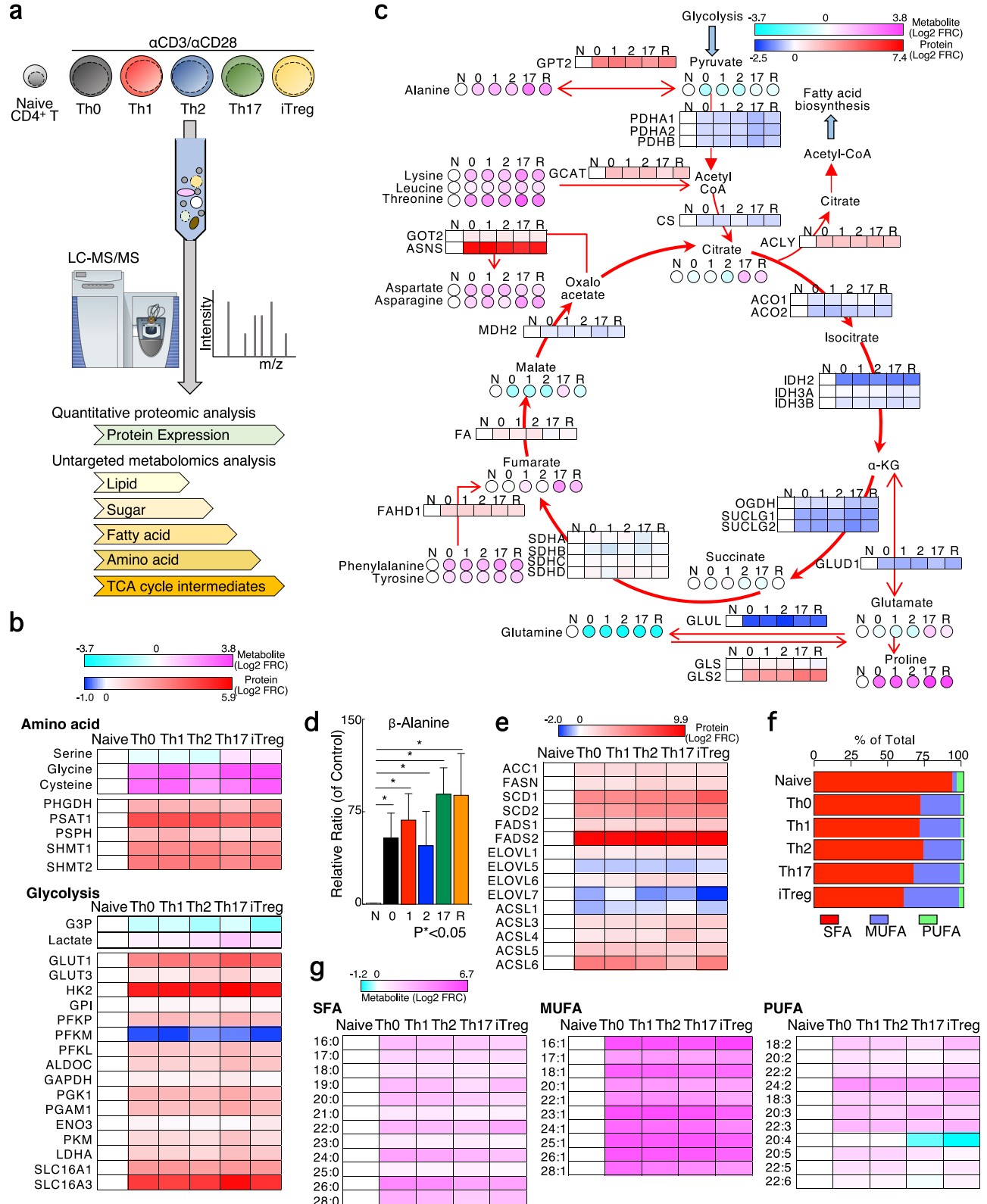

**Fig. 1 | Combination analysis of metabolomic and proteomic data showed T cell activation causes rewiring of the intracellular metabolism. a**, Overview of the experimental design. **b** and **c**, Heatmap depict the metabolites or protein expression related to glycolysis (**b**) or TCA cycle (**c**) in naïve CD4$^+$, Th0, Th1, Th2, Th17, and iTreg cells. **d**, The relative ratio of β-Alanine was shown here. **e**, Heatmap depict the protein expression related to FA biosynthesis in naïve CD4$^+$, Th0, Th1, Th2, Th17, and iTregs cells. **f**, The ratio of fatty acid species detected by lipidomics analysis was shown. **g**, Heatmap depict the levels of fatty acid in naïve CD4$^+$, Th0, Th1, Th2, Th17, and iTregs cells. Four biological replicate was prepared for metabolomics and proteomics analysis. The source data for the figures is provided in Supplementary Data 1 and Supplementary Data 2.

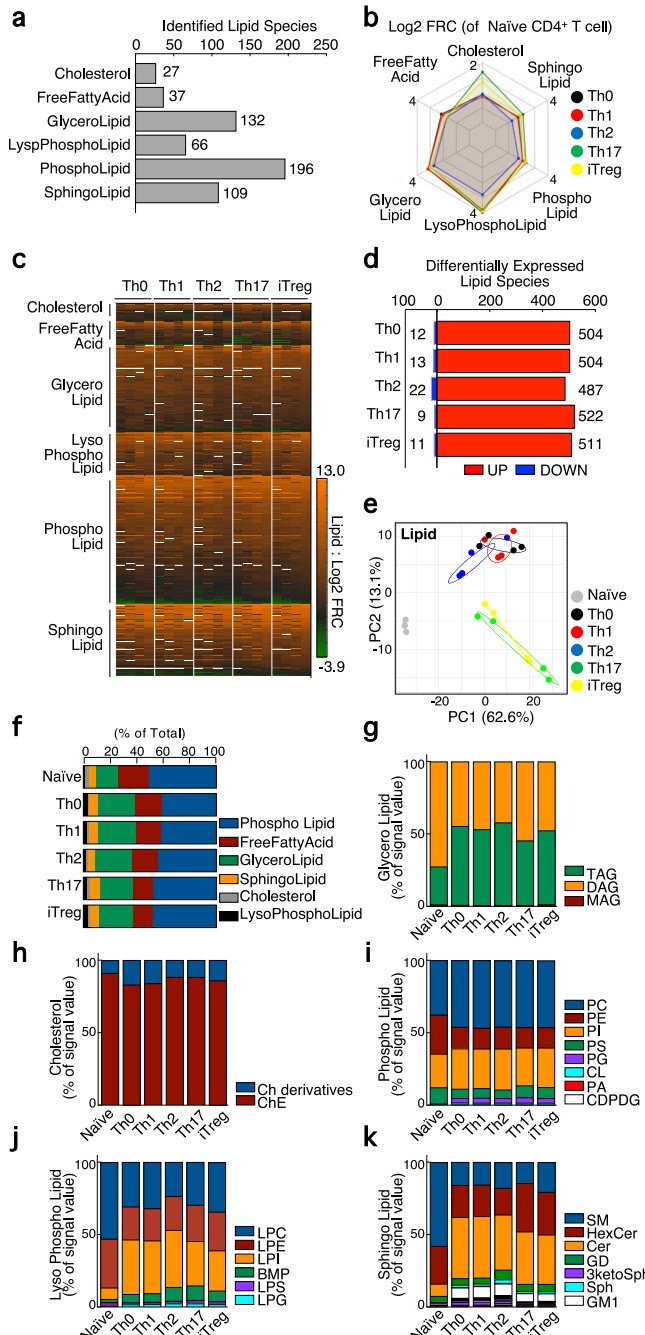

**Fig. 2 | The intracellular lipid metabolism was largely changed by TCR stimulation. a,** The number of detected lipid species by lipidomics analysis was shown. **b,** Spider chart depicts the changes in total amounts of each lipid group compared to naïve CD4+ T cells. **c,** Heatmap depict the levels of each lipid species in Th0, Th1, Th2, Th17, and iTregs cells compared to naïve CD4+ T cells. **d,** The number of differentially expressed lipid species was shown. **e,** PCA depicts lipid profiles of naïve CD4+, Th0, Th1, Th2, Th17, and iTregs cells. **f-k,** Bar plots showed that the ratio of total amounts of each lipid group (**f**), cholesterol (**g**), glycerol lipid (**h**), phospholipid (**i**), lysophospholipid (**j**), or sphingolipid (**k**). Four biological replicate was prepared for metabolomics analysis. The source data for the figures is provided in Supplementary Data 1 and Supplementary Data 2.

## Effector Th cell subsets altered the composition of cellular lipids in comparison to naïve CD4+ T cells

Since activated Th cells dramatically altered fatty acid metabolism, we next analyzed lipidomic data to assess the changes in cellular lipid profiles after TCR stimulation. Our comprehensive cellular lipidomics analysis identified

a total of 567 lipid species. These lipids were further classified based on their structure into cholesterol, free fatty acids, glycerolipids, lysophospholipids, phospholipids, and sphingolipids, with 27, 37, 132, 66, 196, and 109 species, respectively (Fig. 2a). After 48 h TCR stimulation, activated CD4+ T cells exhibited an almost 2-fold increase in cell size and 3-fold increase in the levels of cellular lipid droplets in comparison to the levels detected in naïve CD4+ T cells. (Supplementary Fig. 2a, b). Lipid droplets are composed of monoacylglycerol (MAG), diacylglycerol (DAG), triacylglycerol (TAG), and cholesteryl ester (ChE). When we normalized the lipidomics data based on cell size, the amounts of DAG and ChE were much lower in activated Th cell subsets than the levels in naïve CD4+ T cells. This is not consistent with our results of increased lipid droplets in activated Th cells and a previous study reporting that activated T cells increased intracellular cholesterol contents[21] (Supplementary Fig. 2b). Furthermore, normalizing the data with cell size diminished the changes in the amount of free fatty acids between activated Th cell subsets and naïve T cells. This result is inconsistent with a previous reports suggesting that activated T cells exhibited augmented fatty acid uptake[10]. Considering with these issues, the lipidomics data were based on the cell number when comparing naïve CD4+ T cells and differentiated Th cell subsets. Accordingly, the metabolomics data were also normalized based on the cell number to compare naïve CD4+ T cells and differentiated Th cell subsets. We first observed that activated Th cell increased the overall quantity of these lipid species to over twice that of naïve CD4+ T cells (Fig. 2b). A deeper analysis revealed that the amounts of most lipid species was largely increased in Th cell subsets in comparison to naive CD4+ T cells, the numbers of which were 504, 504, 487, 522, and 511 in Th0, Th1, Th2, Th17, and iTreg cells, respectively (Fig. 2c, d). A principal-component analysis (PCA) demonstrated that TCR-mediated activation had a significant impact on cellular lipid profiles (Fig. 2e). Additionally, Th17 and iTreg cells formed a cluster that was distinct from other activated Th cell clusters including Th0, Th1, and Th2 cells. The stimulation of TCR not only led to the upregulation of lipid biosynthesis, but also resulted in a change in cellular lipid composition as well as free fatty acid levels as shown in Fig. 1g. The TCR stimulation increased the ratio of glycerolipids with values of 15.4% for Naïve CD4+ T cells, 28% for Th0 cells, 28.3% for Th1 cells, 27.1% for Th2 cells, 23.9% for Th17 cells, and 24.7% for iTreg cells (Fig. 2f). In a group of glycerolipid, effector Th cell subsets increased the proportion of TAG, which are incorporated into lipid droplet as energy storage with values of 26.2% for Naïve CD4+ T cells, 54.4% for Th0 cells, 52.3% for Th1 cells, 57.1% for Th2 cells, 44.9% for Th17 cells, and 51.0% for iTreg cell (Fig. 2g). The stimulation of TCR has only a minor impact on the composition of ChE (as depicted in Fig. 2h, with the percentages of 91.2% for Naïve CD4+ T cells, 83.1% for Th0 cells, 84.0% for Th1 cells, 88.3% for Th2 cells, 88.3% for Th17 cells, and 86.0% for iTreg cells). The percentage of phosphatidylcholine (PC) was nearly comparable across each T cell subset (Fig. 2i, with the percentages of 37.5% for Naïve CD4+ T cells, 46.0% for Th0 cells, 46.6% for Th1 cells, 45.8% for Th2 cells, 46.2% for Th17 cells, and 46.3% for iTreg cells). In comparison to naive CD4+ T cells, activated Th cells were found to alter the composition of phosphatidylethanolamine (PE) and phosphatidylserine (PS) from 27.0% to 14-16% or from 11.1% to 6–9%, respectively (Fig. 2i). We also observed substantial changes in the category of lysophospholipids and sphingolipids. Naïve CD4+ T cells exhibited the greatest proportion of LPC, yet TCR stimulation resulted in the most prominent lipid shifting from LPC to LPI. In naïve CD4+ T cells, LPC and LPI comprised 53.0% and 7.83%, respectively (Fig. 2j). In activated Th cell subsets, LPC and LPI occupied 23-34% or 27-39% of total lysophospholipids, respectively. TCR stimulation also largely changed the composition of cellular sphingolipids, with the most abundant lipid shifting from sphingomyelin (SM) to ceramide (Cer). In naïve CD4+ T cells, SM and Cer comprised 58.0% and 8.34%, respectively, while in activated Th cell subsets, SM and Cer occupied 14-20% or 34-43% of total sphingolipids, respectively (Fig. 2k). In addition, TCR-stimulation led to the production of sphingoglycolipids, including ganglioside GM1 (GM1) and ganglioside GD1 (GD1; Fig. 2k). These findings demonstrate that

**Fig. 3 | Activated Th cells enhanced sphingolipid biosynthesis pathway among lipid biosynthesis. a**, Heatmap depict the expression levels of protein related to lipid metabolism. **b**, PCA depicts profiles of protein related to lipid metabolism in naïve CD4$^+$, Th0, Th1, Th2, Th17, and iTregs cells. **c**, Venn diagram showed overlaps and differences between 2.0-fold increased proteins related to lipid metabolism in Th0, Th1, Th2, Th17, or iTregs cells relative to naïve CD4$^+$T cells. **d** and **e**, The lipidomics analysis shows the relative contents of lipid species related to ceramide lipids (**d**) or glycosphingolipid (**e**) in Th0, Th1, Th2, Th17, and iTreg cells compared to naïve CD4$^+$ T cells. Four biological replicate was prepared for metabolomics and proteomics analysis. The source data for the figures is provided in Supplementary Data 1 and Supplementary Data 2.

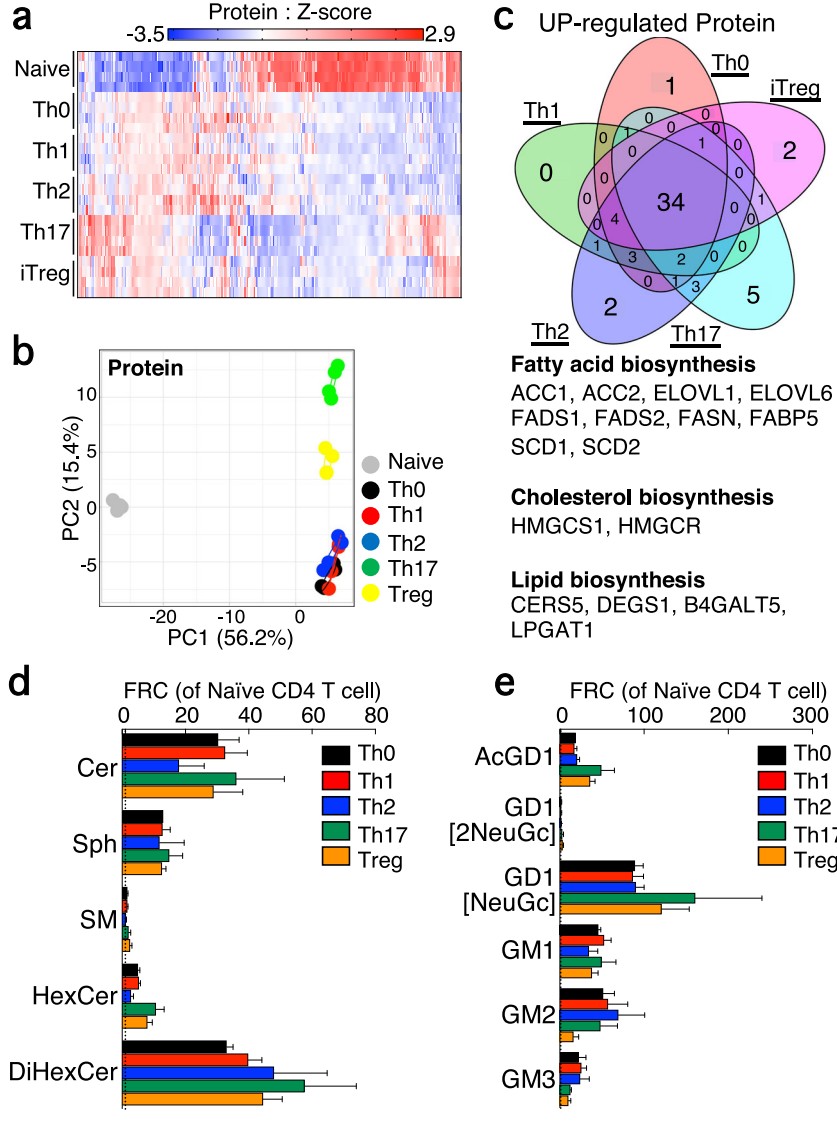

### The activation of a specific Th cell subset resulted in a significant change of metabolites and proteins involved in the biosynthesis pathway of sphingolipids and glycosphingolipids

Subsequently, we seek to determine the species of lipids that influence the generation of effector Th cell subsets. The activated Th cell subsets differ in cell size; however, the differences in cell size were comparable under our experimental conditions (Supplementary Fig. 2a, b). Thus, to accurately compare lipid contents, the data were normalized by the signal intensity of total lipid species when comparing the lipidomics data of each Th cell subset to Th0 cells (Formula: the signal value of each lipid metabolite multiplied by 2000000/total signal value of lipid metabolites). To this end, we conducted an initial evaluation of the expression of 269 proteins involved in lipid metabolism in naïve CD4$^+$ T cells, Th0, Th1, Th2, Th17, and iTreg cells. Our proteomic analysis revealed that protein expression levels related to lipid metabolism underwent dynamic changes following TCR stimulation (Fig. 3a, b). As indicated by the PCA plots of lipid metabolites (Fig. 2e), Th0, Th1, and Th2 cells displayed similar protein profiles, whereas Th17 and iTreg cells exhibited greater similarity to each other than to the other cell states within the PCA space. We then sought to identify proteins that were

commonly changed among the various T cell subsets. Our observations indicated that the expression of diacylglycerol kinase (DGKA, DGKB, DGKZ), phosphatidylinositol kinase (PIK3CB, PIK3CD, PIP4K2A, PIP4K2B), and phospholipase (PLBD1, PLCB2, PLCB3, PLCL1) was decreased in a common manner among effector Th cell subsets (Supplementary Fig. 2c). Each Th cell subset commonly upregulated proteins that are essential for the biosynthesis of fatty acids (ACC1, FASN), elongation of fatty acids (ELOVL1, ELOVL6), desaturation of fatty acids (FADS1, FADS2, SCD2), fatty acid binding protein (FABP5), and cholesterol synthesis (HMGCS1 and HMGCR) (Fig. 3c). Especially, the proteins predominantly associated with the biosynthesis of lipids, primarily sphingolipids, that were commonly upregulated include CERS5, DEGS1, and B4GALT5 (Fig. 3c). In accordance with upregulation of these proteins, the quantity of ceramides (Cer) was considerably increased in effector T cell subsets (Fig. 3d). Lipidomics analysis further revealed substantial elevations in the amounts of cardiolipin (CL) and phosphatidylglycerol (PG) in glycerophospholipid, bis monoacylglycero phosphate (BMP), LPI, and LPG in lysophospholipid, TAG in glycerolipid (Supplementary Fig. 2d–f). Nevertheless, the synthases of these lipids were not among the commonly altered proteins in effector Th cell subsets except for LPGAT1. Thus, we next focused on the sphingolipid synthesis pathway for the following reasons: Proteomic analysis indicated that proteins related to the sphingolipid synthesis pathway were upregulated in a general manner (Supplementary Fig. 2g), and lipidomic

analysis showed substantial increase in the quantity of Cer and significant changes in the composition of sphingolipids (Fig. 2k). Subsequently, we conducted a detailed evaluation of the changes in sphingolipids during T-cell activation. We first observed that the amounts of ceramide lipids and their precursor, sphinganine, was greatly increased (Fig. 3d). Cer serves as the hub of sphingolipid metabolism and is responsible for the generation of sphingomyelin, sphingosine, and glycosphingolipids. Although the amounts of SM remained largely unchanged, there were increases in the amounts of hexosyl Cer (HexCer) and di-hexosyl Cer (DiHexCer), which are types of glycosphingolipids (Fig. 3d). In addition, although many gangliosides, which are metabolites of these glycosphingolipids, were barely detectable in naive CD4$^+$ T cells, effector Th cell subsets increased the levels of ganglioside, such as GM2, GM3, and GT1a (Fig. 3e). Taken together, these data suggest that TCR activation enhances the utilization of the sphingolipid synthesis pathway and confers the regulation of glycosphingolipid metabolism.

### Activated sphingolipid metabolism is required for the proper generation of Th17 and iTreg cells

Next, to explore the influence of sphingolipid biosynthesis on the differentiation of effector Th cells, we inhibited sphingolipid production through the utilization of myriocin, a molecule known to inhibit the initial steps of sphingolipid biosynthesis. Inhibition of sphingolipid biosynthesis by myriocin resulted in a moderate decrease in the number of Th17 and iTreg cells (Fig. 4a). FACS analysis also showed that myriocin-treated Th17 and iTreg cells showed impaired TCR-induced cell division with slight changes in the cell death rate (Supplementary Fig. 3a–d). We also observed that the myriocin-treated Th17 and iTreg cells substantially reduced the amounts of Cer and its downstream glycosphingolipids, including HexCer, DiHexCer, GD1, and AcGD1 (Fig. 4b, c and Supplementary Fig. 3e, f). Subsequently, the effect of myriocin on cell cycle progression was analyzed *via* the evaluation of BrdU incorporation in Th17 and iTreg cells. As a consequence, myriocin treatment prevented Th17 and iTreg cells from entering the S-phase following TCR stimulation (Fig. 4d, e). No significant changes were observed in the number and cytokine production of Th0, Th1, and Th2 cells (Supplementary Fig. 3g–i). Since TGFβ is required to induce Th17 and iTreg differentiation, we next tested whether TGFβ changes the sensitivity to myriocin in Th1 and Th2 cells. We first found that in the presence of TGFβ, myriocin treatment caused a decrease in the number of Th1 and Th2 cells (Supplementary Fig. 3j). However, there were no significant changes in the production of IFNγ and IL-9, which were produced by TGFβ-treated Th2 cells defined as Th9 cells (Supplementary Fig. 3k, l). In addition to impaired cell division, treatment of myriocin resulted in the inhibition of the cytokine production of Th17 cell (Fig. 4f, g and Supplementary Fig. 4a, b). We also addressed whether myriocin treatment suppresses the expression levels of FOXP3, IL-2, and IL-10. Our findings also indicated that the inhibition of sphingolipid biosynthesis led to a decrease in PD-1 expression and IL-2 production, but there were no significant changes in the expression of FOXP3 and IL-10 at either the mRNA or protein level (Fig. 4h, i and Supplementary Fig. 4b–d). Although treatment with myriocin from 48 h after T cell activation failed to suppress differentiation of Th17 or iTreg cells, treatment with myriocin 24 h after TCR stimulation suppressed IL-17A production or the PD-1 expression (Supplementary Fig. 4e–h). Consistent with decreased PD-1 expression and IL-2 production, myriocin-treated iTreg cells showed impaired suppressive capacity (Supplementary Fig. 4i, j). Next, we investigated whether supplementation with sphingolipid restored the effect of myriocin on Th17 and iTreg cell differentiation. Although the number of myriocin-treated iTreg cells was almost unchanged by SM, Sph, Cer (d18:1/16:0), and Cer (d18:1/18:0) treatment (Supplementary Fig. 4k, l), Sph and Cer (d18:1/16:0) moderately recovered the PD1 expression (Fig. 4j, k). We also found that the supplementation with SM, Sph, Cer (d18:1/16:0), and Cer (d18:1/18:0) moderately restored cell number of myriocin-treated Th17 cells. Supplementation with these lipid species did not affect IL-17A and IL-17F production (Fig. 4l, m and Supplementary Fig. 4m, p).

We next performed gene edition of *Sptlc1* and *Sptlc2*, which are key targets of myriocin, to elucidate the importance of sphingolipid metabolism for Th17 and iTreg differentiation (Supplementary Fig. 5a). The genetic deletion of *Sptlc1* and *Sptlc2* moderately suppressed the differentiation of Th17 cells and the expression of PD-1 and IL-2 in iTreg cells (Fig. 4n–q and Supplementary Fig. 5b–e). Genetic deletion of *Sptlc1* and *Sptlc2* caused impaired production of sphingolipid and glycosphingolipid in Th17 and iTreg cells (Supplementary Fig. 5f–i). Collectively, our combined analysis demonstrates that effector Th cell subsets exhibit enhanced sphingolipid and glycosphingolipid metabolism. Additionally, the generation of both Th17 and iTreg cells was inhibited by the ceramide biosynthesis inhibitor.

### Glycosphingolipid metabolism was facilitated in Th17 and iTreg cells in comparison to Th0, Th1, and Th2 cells

The inhibition of sphingolipid biosynthesis selectively suppressed the generation of Th17 and iTreg cells, suggesting characteristic differences in cellular lipid metabolism between these effector Th cell subsets and Th0, Th1, and Th2 cells. To further investigate this cellar metabolic profile, we examined global lipidomic profiling and analyzed protein expression related to lipid metabolism among the various effector Th cell subsets. Results from PCA revealed that Th0, Th1, and Th2 cells showed similar lipid and protein profiles. While Th17 and iTreg cells were more alike to each other than to the other cell states in PCA space (Fig. 5a, b). The analysis also revealed elevated amounts of sphingolipids in Th17 and iTreg cells in comparison to non-polarizing Th0 cells, as well as decreased levels of 3-ketosphinganine, an intermediate in sphingolipid synthesis, in Th17 and iTreg cells (Fig. 5c, d). These results led us to focus on sphingolipids and their downstream metabolic pathways, where we observed increased levels of HexCer, DiHexCer, GD1, and GD1[O-acetyl NeuAc] (AcGD1) (Fig. 5e). Although total amounts of ceramide lipid levels did not significantly differ in Th17 and iTreg cells in comparison to Th0 cells, a closer examination revealed that the proportion of individual ceramide species in the most of HexCer, DiHexCer, GD1, and AcGD1 was altered in Th17 and iTreg cells (Fig. 5f–h). Furthermore, inhibition of glycosphingolipid through pharmacological inhibitor, GENZ-123346, suppressed the number of Th17 and iTreg cells as well as myriocin treatment (Fig. 5i–k and Supplementary Fig. 6a, b). In accordance with these results, GENZ-123346 inhibited the cytokine production of Th17 cells and PD-1 expression on Treg cells, further demonstrating the importance of the glycospingolipid biosynthesis pathway in the differentiation of Th17 and iTreg cells (Fig. 5l, m and Supplementary Fig. 6c, d). GENZ-123346 prevent the conversion of Cer to HexCer. Consequently, treatment of GENZ-123346 with Th17 and iTreg cells caused accumulation of Cer and depletion of glycosphingolipid, including HexCer, DiHexCer, GD1, and AcGD1 (Supplementary Fig. 6e–h). These data indicated that glycosphingolipid metabolism is required to regulate Th17 and iTreg cells differentiation.

It has also been reported there was differences in the metabolic profiles and requirement of pathogenic Th17 cells and non-pathogenic Th17 cells[22]. Therefore, we differentiated naïve CD4$^+$ T cells into pathogenic (p) Th17 cells in the presence of IL-1β, IL-6, and IL-23, and evaluated whether the inhibition of sphingolipid or glycosphingolipid metabolism suppresses the generation of pTh17 cells. Although we found that the number of pathogenic Th17 cells decreased with the addition of myriocin or GENZ-123346, the production of cytokines was not substantially affected under these conditions (Supplementary Fig. 6i–k). Previous studies have indicated that glycosphingolipids and cholesterol may affect early TCR signaling, we evaluated the effects of myriocin and GENZ-123346 on phosphorylation of proteins downstream TCR signaling[23]. After 30 min TCR stimulation, inhibition of sphingolipid or glycosphingolipid metabolism by myriocin or GENZ-123346 in Th17 or iTreg cells did not substantially affect the phosphorylation of Src, Lck, and Lat (Supplementary Fig. 6I, m). We also investigated whether myriocin inhibited Th17 and iTreg cell differentiation after early TCR stimulation. Although treatment with myriocin from 48 h after T cell

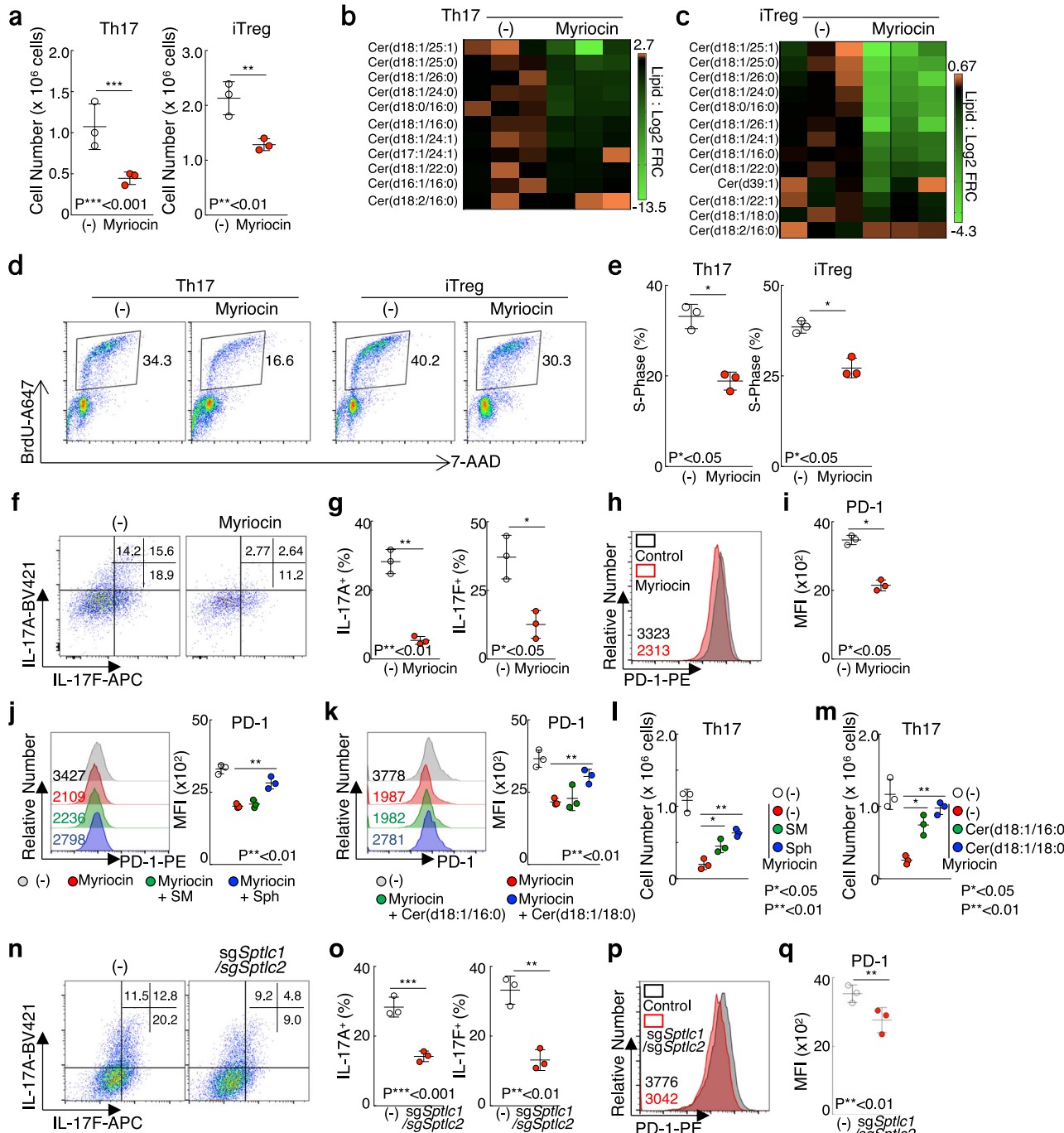

**Fig. 4 | Inhibition of ceramide biosynthesis resulted in the suppression of Th17 and iTreg differentiation. a**, Cell number of Th17 or iTreg cells treated with DMSO or 6.25 µM myriocin was shown. **b,c** Heatmap depict the levels of ceramide in myriocin-treated Th17 (**b**) and iTreg cells (**c**). **d**, BrdU incorporation by Th17 or iTreg cells treated with myriocin was examined by flow cytometry. **e**, Summary data of cell cycle related to (**d**) was shown. **f**, Intracellular staining of IL-17A and IL-17F in Th17 cells treated with myriocin was shown. **g**, Summary data of IL-17A and IL-17F expression related to (**f**) was shown. **h**, Surface staining of PD-1 in iTreg cells treated with myriocin was shown. **i**, Summary data of PD-1 expression related to iTreg cells was shown. **j, k**, FACS analysis shows PD-1 expression of iTreg cells treated with

2 µM SM or 2 µM Sph (**j**), 0.2 µM Cer(d18:1/16:0) or 0.2 µM Cer(d18:1/18:0) (**k**) in the presence of 6.25 µM myriocin. **l, m**, FACS analysis shows the cell number of Th17 cells treated with 2 µM SM or 2 µM Sph (**l**), 0.2 µM Cer(d18:1/16:0) or 0.2 µM Cer(d18:1/18:0) (**m**) in the presence of 6.25 µM myriocin. **n**, Intracellular staining of IL-17A and IL-17F in sg*Sptlc1*/sg*Sptlc2* Th17 cells was shown. **o**, Summary data of IL-17A and IL-17F expression related to (**n**) was shown. **p**, Surface staining of PD-1 in sg*Sptlc1*/sg*Sptlc2* iTreg cells treated was shown. **q**, Summary data of PD-1 expression related to (**p**) was shown. Three independent experiments were performed and showed similar results. Error bar indicates SD. The source data for the figures is provided in Supplementary Data 1 and Supplementary Data 2.

activation failed to suppress differentiation of Th17 or iTreg cells, myriocin treatment with 24 h after TCR stimulation suppressed IL-17A production or PD-1 expression (Supplementary Fig. 4e–h). Taken together, impaired Th17 and iTreg cell differentiation did not appear to be highly dependent on TCR stimulation.

## Methods
### Mice
C57BL/6 mice were purchased from CLEA Japan. All mice were used at 6–8 weeks old and were maintained under specific-pathogen-free conditions. Almost equal number of male and female animal was used for this

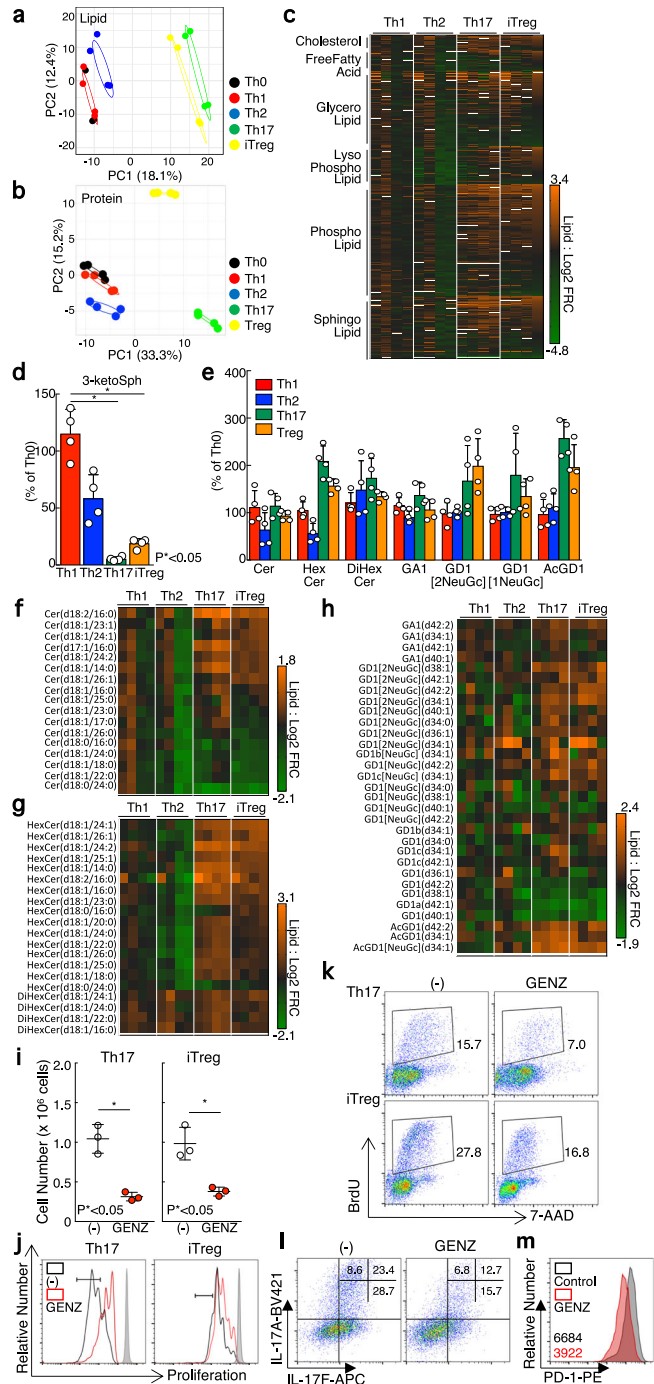

**Fig. 5 | Glycosphingolipid biosynthesis pathway is selectively upregulated in Th17 and iTreg cells compared to Th0, Th1 and Th2 cells. a, b** PCA depicts profiles of lipid (**a**) or protein related to lipid metabolism (**b**) in Th0, Th1, Th2, Th17, and iTregs cells. **c**, Heatmap depicts the expression levels of each lipid species in Th1, Th2, Th17, and iTreg cells compared to Th0 cells. **d**, Relative levels of 3-ketosphinanine was shown in Th1, Th2, Th17, and iTreg cells compared to Th0 cells. **e**, Lipidomics analysis shows the relative contents of lipid species related to ceramide glycosphingolipid in Th1, Th2, Th17, and iTreg cells compared to Th0 cells. **f-h**, Heatmap depicts the expression levels of each ceramide species (**f**), each glycol-ceramide species (**g**), or each ganglioside species (**h**) was shown in Th1, Th2, Th17, and iTreg cells compared to Th0 cells. **i**, Cell number of Th17 or iTreg cells treated with DMSO or 2.5 μM GENZ-123346 was shown. **j**, Naïve CD4+ T cells were labelled with e670 proliferation dye and stimulated with immobilized anti-TCRβ mAb and anti-CD28 mAb in the presence of GENZ-123346 under Th17 or iTreg polarization condition. **k**, BrdU incorporation by Th17 or iTreg cells treated with GENZ-123346 was examined by flow cytometry. **l**, Intracellular staining of IL-17A and IL-17F in Th17 cells treated with myriocin was shown. **m**, Surface staining of PD-1 in iTreg cells treated with GENZ-123346 was shown. Four biological replicate was prepared for metabolomics and proteomics analysis. Three independent experiments were performed and showed similar results (**i-m**). Error bar indicates SD. The source data for the figures is provided in Supplementary Data 1 and Supplementary Data 2.

1 mM sodium pyruvate, 2 mM L-glutamine, and 10% U.S. origin FBS. The culture medium contained 1 μg/ml anti-CD28 antibody (clone 37.51, BioLegend) and cytokine that induce differentiation of Th cell subsets. Th0 cell cultures contained 15 ng/ml IL-2 (WAKO), 1 μg/ml anti-IL-4 antibody (BioLegend), and 1 μg/ml anti-IFNγ antibody (BioLegend). Th1 cell cultures contained 15 ng/ml IL-2, 10 ng/ml recombinant mouse IL-12 (WAKO) and 1 μg/ml anti-IL-4 antibody. Th2 cell cultures contained 15 ng/ml IL-2, recombinant mouse 10 ng/ml IL-4 (WAKO), and 1 μg/ml anti-IFNγ antibody. Th17 cell cultures contained 10 ng/ml IL-6 (PeproTech 216-16), 1 ng/ml TGFβ (PeproTech 100-21 C), 1 μg/ml anti-IL-2 antibody (BioLegend), 1 μg/ml anti-IL-4 antibody, and 1 μg/ml anti-IFNγ antibody. Regulatory T cell cultures contained 30 ng/ml IL-2, 10 ng/ml TGFβ, 1 μg/ml anti-IL-4 antibody, and 1 μg/ml anti-IFNγ antibody. 6.25 μM myriocin, 2.5 μM GENZ-123346, 2 μM SM, 2 μM Sph, 0.2 μM Cer (d18:1/16:0) or 0.2 μM Cer (d18:1/18:0) was treated under the indicated Th cell subsets polarization conditions.

## Meta-analysis of proteomics Data

The data was obtained from our previous reports deposited to the ProteomeXchange Consortium via the jPOST partner repository (http://www.proteomexchange.org/) with the dataset identifier PXD036065[24]. The value of protein intensity was transformed to log2, and then each protein was filtered to contain more than 70% valid value in at least one group. The remaining missing values were imputed by random numbers drawn from a normal distribution (width, 0.3; downshift, 2.8) in Perseus v1.6.15.0[25]. 2.0-fold changed genes was defined as differentially expressed proteins. PCA analysis and heatmap were depicted with R software (https://cran.r-project.org/) (v 3.6.0).

## Metabolome analysis

The samples, including naïve CD4+ T, Th0, Th1, Th2, Th17, and iTreg cells, were prepared as described in "Cell preparation". After dead cells were excluded using Dead Cell Removal Kit (Miltenyi Biotec #130-090-101), $2.5 \times 10^6$ cells of each sample were collected and washed with PBS (-) twice. After removal of supernatant, the pellet was frozen at −70 °C.

Targeted metabolomics was performed by gas chromatograph-mass spectrometer (GC-MS) with methoxime and trimethylsilyl derivatization methods[26]. The frozen samples ($2.5 \times 10^6$ cells) were suspended in a solvent mixture (475 μl) of water, methanol and chloroform (1: 2.5: 1.25) and mixed by vortex mixer. Subsequently, 125 μl water and 125 μl chloroform were added to the suspension. After centrifugation at 20,400×*g* for 10 min, 100 μl of the upper layer was collected and dried under N2-gas flow. The dried samples were dissolved in 40 μl of 20 mg/ml methoxyamine hydrochloride

study. The animal experiments were performed with protocols approved by the Institution Animal Care and Use Committee of KAZUSA DNA research institute (Registration number: 30-1-002). Experiments and animal care were performed according to the guidelines of Kazusa DNA Research Institute.

## Cell preparation

Splenic naïve CD4+ T cells were obtained by the negative selection using the Mojo Sort Mouse CD4 T Cell Isolation Kit (Biolegend #480006) and positive selection using CD62L MicroBeads, mouse (Miltenyi Biotec #130-049-701). Naïve CD4+ T cells were plated onto 24-well tissue culture plates (Costar #3526) pre-coated with 10 mg/ml anti-TCRβ antibody (H57-597, BioLegend) for 2 days. RPMI 1640 medium supplemented with 55 μM 2-mercaptoethanol, 10 μM non-essential amino acids solution, 10 mM HEPES,

dissolved in pyridine and incubated at 30 °C for 90 min. Finally, 40 µl of *N*-methyl-*N*-(trimethylsilyl)-trifluoroacetamide was added to the solution and the solution was incubated further at 37 °C for 30 min. The derivatized samples were analyzed by GC-MS (GCMS QP-2010 Ultra; Shimadzu Corporation) equipped with an Agilent DB-5 column (0.25 mm × 30 m, 1-µm-film thickness; Agilent).

## Lipidome analysis

Untargeted lipidomics was performed as reported previously with some modifications[27]. Briefly, the frozen pellets (2.5×10^6 cells) were re-dissolved in 150 µl of chloroform:methanol (1:2) containing EquiSPLASH (Avanti Polar Lipids) for internal standards. After sonication for 30 sec, 10 µL of water was added and vigorously agitated for at 750 rpm for 20 min at 20 °C. The supernatant was collected by centrifugation at 1670 g for 10 min at 20 °C and transferred to a LC vial. LC-MS/MS analysis was carried out using a quadruple time-of-flight (Q TOF)/MS (TripleTOF 6600; SCIEX) coupled with an ACQUITY UPLC system (Waters). The LC separation was performed with gradient elution of mobile phase (A) [methanol/acetonitrile/water (1:1:3, v/v/v) containing 5 mM ammonium acetate (Wako Chemicals) and 10 nM EDTA (Dojindo)] and mobile phase (B) [isopropanol (Wako Chemicals) containing 5 mM ammonium acetate and 10 nM EDTA]. The flow rate was 300 µl/min at 45 °C using a L-column3 C18 (50 × 2.0 mm i.d., particle size 2.0 µm; Chemicals Evaluation and Research Institute). The solvent composition started at 100% (A) for the first 1 min and was changed linearly to 64% (B) at 7.5 min, where it was held for 4.5 min. The gradient was increased linearly to 82.5% (B) at 12.5 min, followed by 85% (B) at 19 min, 95% (B) at 20 min, 100% (A) at 20.1 min, and 100% (A) at 25 min. The raw data files from Q TOF/MS were converted to MGF files using the program of SCIEX MS converter for the quantitative analysis with 2DICAL (Mitsui Knowledge Industry). Identification of the molecular species was accomplished by comparison with retention times and MS/MS spectra data from the information-dependent acquisition (IDA) mode[28].

## Cas9 mediated-genome-editing

The short guide RNA was designed using the online tool provided by CHOPCHOP[29]. Freshly isolated naïve CD4 T cells were activated with plate bound anti-CD3 and CD28 antibodies. Cas9 proteins were prepared immediately before experiments by incubating 1 µg Cas9 with 0.3 µg sgRNA in transfection buffer at room temperature for 10 min. 24 h after T cell activation, these cells were electroporated with a Neon transfection kit and device (Thermo).

Target sequences used in this study are shown below.
sg*Sptlc1* : 5′- TTTGTCGTAGAATCCTCGCAAGG-3′
sg*Sptlc2* : 5′- GCGGAACATTGGTGTAGTTGTGG-3′

## Quantitative real-time PCR

Total RNA was isolated with the TRIzol reagent (Invitrogen #15596-018). cDNA was synthesized with an oligo (dT) primer and Superscript II RT (Invitrogen #18064-014). Quantitative RT-PCR was performed using TB Green Real Time PCR kit (Takara #RR820A). Primers were purchased from Thermo Fisher Scientific. Gene expression was normalized with the Hprt mRNA signal or the 18S ribosomal RNA signal. Primer sequences used in this study are shown below.

*18S*_FW : 5′-AAATCAGTTATGGTTCCTTTGGTC-3′
*18S*_RV : 5′-GCTCTAGAATTACCACAGTTATCCAA-3′
*Hprt*_FW : 5′-TCCTCCTCAGACCGCTTTT-3′
*Hprt*_RV : 5′-CCTGGTTCATCATCGCTAATC-3′
*Il-17A*_FW : 5′-CAGGGAGAGCTTCATCTGTGT-3′
*Il-17A*_RV : 5′-GCTGAGCTTTGAGGGATGAT-3
*Il-17F*_FW : 5′-CCCAGGAAGACATACTTAGAAGAAA-3′
*Il-17F*_RV : 5′-CAACAGTAGCAAAGACTTGACCA-3

## FACS analysis

For surface staining, anti-PD-L1 PE (1:200, J43, BD Biosciences #551892) was stained for 30 min on ice and dead cell was stained with Propidium iodide (1:2000, DOJINDO #341-07881) before FACS analysis. For intracellular staining, dead cells were first stained with Fixable Viability Dye eFluor 780 (1:1000, eBioscience #65-0865-14) for 10 min. For cytokine staining, sample preparation was conducted with 4% PFA for 10 min at 4 °C and incubated with permeabilization buffer for 10 min on ice (50 mM NaCl, 5 mM EDTA, 0.5% Triton-X). After incubating with blocking buffer for 15 min on ice (0.3% BSA), the cells were stained with anti-TNFα PE (1:200, MP6-XT22, Biolegend# 506306), anti-IFNγ FITC (1:50, XMG1.2, BD Biosciences #554411), anti-IL-2 APC (1:1000, JES6-5H4, BD Biosciences #554429), anti-IL-4 BV421 (1:200, 11B11, BD Biosciences #562915), anti-IL-17A BV421 (1:200, TC11-18H10.1, Biolegend#506925), anti-IL-17F APC (1:200, O79-289, BD Biosciences #561630), or for 30 min in the dark. Cell proliferation and cell cycle were analyzed with Proliferation dye e670 (Invitrogen #65-0840-90) or BrdU Flow kit (BD Biosciences #557891) as according to the manufacture's protocol. Flow cytometric data were analyzed after removal of dead cells and doublets cells with Flowjo software (version 10.4).

## Statistics and Reproducibility

Data are expressed as mean ± SD. The data were analyzed with the Graphpad Prism software program (version 7). Differences were assessed using unpaired two-tailed student t tests or one-way ANOVA followed by tukey's multiple comparisons test. Differences with P values of <0.05 were considered to be statistically significant. No data were excluded from the analysis of experiments. Mice were commercially sourced and randomized into experimental groups upon arrival, and all animals within a single experiment were processed at the same time. For metabolomics, the investigator was blinded. Data display similar variance between groups and are normally distributed where parametric tests are used.

## Reporting summary

Further information on research design is available in the Nature Portfolio Reporting Summary linked to this article.

## Discussion

In this study, we performed a comprehensive analysis to investigate the impact of TCR and cytokine stimulation on intracellular metabolism in terms of protein and metabolite levels. Our multi-omics analysis revealed that activated Th cells significantly enhance amino acid, glucose, TCA cycle, and fatty acid metabolic pathway, as previously reported[6]. Especially, there was a marked change in the quantity of β-alanine, which serves as a precursor for acetyl-CoA, upon T cell activation. Effector Th cells dramatically increased the production of SFA, MUFA, and PUFA, and modify FA composition. Although most lipid production was elevated, TCR stimulation particularly heightened sphingolipid biosynthesis. The profiles of lipid metabolites and proteins related to lipid metabolism in Th17 and iTreg cells were significantly different from that in Th0, Th1, and Th2 cells. The inhibition of sphingolipid and glycosphingolipid biosynthesis pathways arrested the cell cycle and cell proliferation of Th17 and iTreg cells, but not Th0, Th1, and Th2 cells. Furthermore, inhibition of sphingolipid biosynthesis led to the suppression of cytokine production of Th17 cells and PD-1 expression on iTreg cells. Thus, our research provides notable insights into the utility of our combined metabolomics and proteomics analysis in furthering the understanding of metabolic transition during Th cell differentiation.

TCR stimulation exhibited a significant impact on the sphingolipid biosynthesis pathway. Especially, activated Th cells substantially elevated the levels of Cer, which acts as a key hub in sphingolipid metabolism, leading to the generation of sphingomyelin, sphingosine, and glycosphingolipids. Our proteomic analysis revealed that changes in the protein expression of CerS family upon TCR stimulation. Specifically, we observed almost 2-fold increase in CerS2 and Cers5 expression in activated Th cells, which synthesize C22-/C24-/C26-Cer or C14-/C16-Cer, respectively[30,31]. In contrast, only slight changes were observed in the expression of CerS4 and CerS6, which produce C18-/C20-Cer or C14-/C16-Cer, respectively[30,31]. Consistent

with the upregulation of CerS2, the proportion of Cer (d18:1/24:0) in ceramide lipids was significantly increased following TCR stimulation (from 8.19% in naïve CD4$^+$ T cells to 26.9% in Th0 cells, 26.2% in Th1 cells, 21.4% in Th2 cells, 17.5% in Th17 cells, and 16.8% in iTreg cells). Furthermore, we found that in naïve CD4$^+$ T cells, HexCer (d18:1/24:0) and DiHexCer (d18:1/24:0) accounted for 9.9% and 0.0%, respectively, whereas in activated Th cell subsets, HexCer (d18:1/24:0) and DiHexCer (d18:1/24:0) occupied 29-39% or 63-73% of theses lipids, respectively. These results suggest that activated Th cells upregulate CerS2 expression to produce Cer (d18:1/24:0) for glycosphingolipid biosynthesis, including HexCer (d18:1/24:0) and DiHexCer (d18:1/24:0). Importantly, a previous study reported that mice deficient in CerS2 exhibited reduced sensitivity to ovalbumin-induced allergic asthma, accompanied by impaired Th2 responses[32]. Upon antigen stimulation, the TCR aggregates into plasma membrane domains known as lipid rafts, which primarily comprise sphingolipids, glycosphingolipids, cholesterol, and phospholipids. The impairment of downstream TCR signaling in CerS2 or CerS6 deficient CD4$^+$ T cells highlights the importance of Cer for the regulation of TCR stimulation[31,32]. However, detailed investigations are necessary to elucidate the specific roles of Cer, HexCer, and DiHexCer in the regulation of TCR stimulation.

We have previously reported that the PPARγ-dependent FA uptake program is essential for a rapid proliferation of CD4$^+$ T cells[10]. In the current study, we also observed substantial changes in the fatty acid biosynthesis pathway following TCR stimulation. Activated Th cells exhibited an increase in the amounts of SFA, MUFA、and PUFA, thus leading to modifications in the desaturation status of fatty acids. Importantly, in consistent with the substantial upregulation of SCD2, there was a marked augmentation in the amounts of MUFAs. Among the MUFAs, TCR stimulation exhibited large impact on the level of oleic acid with values of 1.83% for Naïve CD4$^+$ T cells, 19.7% for Th0 cells, 20.0% for Th1 cells, 16.6% for Th2 cells, 20.3% for Th17 cells, and 24.7% for iTreg cells. MUFA metabolism is required to maintain mitochondrial metabolism and mTOR activity, avoiding excessive autophagy and ER stress in proliferating B cells[33]. MUFA metabolism is also necessary for triggering the type I-IFN response, thereby activating anti-viral responses[34]. This report showed that Th1 cell exhibited the greatest proportion of oleic acids (16.8%) as major components of cellular lipids. Gene deletion of *Scd2* decreases the amounts of oleic acid containing TAG, resulting in the I-IFN production. The supplementation of oleic acid suppresses the type I interferon response in *Scd2*-deficient Th1 cells, suggesting the importance of oleic acid in anti-viral responses.

The profiles of lipid metabolites in Th17 and iTregs was far different from those in Th0, Th1, and Th2 cells, particularly with regard to the activation of sphingolipid and glycosphingolipid biosynthesis pathways. A recent study demonstrated that liver X receptor (LXR) regulates genes associated with glycosphingolipid biosynthesis in primary human T cells. Thus, in turn, LXR affects T cell plasma membrane lipid composition, subsequent immune synapse formation, and TCR-mediated signaling[23]. Activation of LXR by pharmacological means in this study increased the amounts of glycosphingolipids by directly binding to the *Ugcg* locus, which encodes transcripts of UDP-glucose ceramide glucosyltransferase (GCS). Full activation of LXR in mouse embryonic fibroblasts requires TGFβ, which recruits a complex containing RAP250 that can interact with both LXR/RXR and Smad2/3[35]. TGFβ, a cytokine commonly used to induce both Th17 and iTreg cell differentiation, may therefore contribute to the profiles of sphingolipids and glycosphingolipids in these subsets. Thus, it could be possible that the TGFβ-RAP250-LXR axis contributes to the construct unique lipid profiles in Th17 and iTreg cells. It has been also reported that aberrant TGFβ signaling contributes to dysregulation of sphingolipid metabolism in intrauterine growth restriction (IUGR)[36]. TGFβ-mediacted Alk5/Smad2 or Alk1/Smad1 pathway contributes to the disruption of sphingolipid metabolism during IUGR. Alk5/Smad2 contributes to the Cer breakdown, resulting in the accumulation of downstream glycosphingolipids. In addition, impaired TGFβ signaling *via* Alk1/Smad1 contributed to the sphingosine buildup in IUGR. Although several studies have

reported the relationship between sphingolipid metabolism and TGFβ signaling, further study is needed to investigate how sphingolipid metabolism is regulated by TGFβ signaling.

The inhibition of sphingolipid or glycosphingolipid pathway has been shown to suppress the proliferation and differentiation of Th17 and iTreg cells, but not Th0, Th1, and Th2 cells. Treatment of myriocin or GENZ-123346, or gene ablation of *Sptlc1/2* decreased the amounts of glycosphingolipid, including HexCer, DiHexCer, GD1, and AcGD1. Thus, we consider that glycosphingolipid, a downstream lipid of Cer is required for proper Th17 and iTreg differentiation. Supplementation of Sph or Cer(d18:1/18:0) recovered PD-1 expression on myriocin-treated iTreg cells and SM, Sph, Cer(d18:1/16:0), or Cer(d18:1:18:0) moderately restored cell number of myriocin-treated Th17 cells. These results are consistent with the observation that human differentiated Th17 and iTreg cells increased the levels of sphingolipid and glycosphingolipid, and contained different composition of these lipids among CD4$^+$ Th cell subsets[18]. Taken together, our data suggested that Th17 cells and iTreg cells differentially utilize sphingolipids in the regulation of cell proliferation and marker protein expression. Several studies have explored the role of sphingolipid biosynthesis pathway in regulating specific Th cell subsets[37–39]. Inhibition of acid sphingomyelinase (Asm) has also been demonstrated to suppress Th17 cell differentiation[37]. Another study reported that gene deletion of *Asm* in T cells led to the enhancement of Treg cells both in vivo and in vitro settings[38]. Ceramide-mediated activation of PP2A endows Treg cells with the necessary phosphatase activity to control mTORC1 and establish their distinctive tolerogenic metabolic and cytokine profiles[39]. As previously reported, ORMDL3 is known to negatively regulate the rate limiting enzyme of sphingolipid biosynthesis, SPT. A cohort of 17q21 risk SNP carriers showed upregulation of locus-related gene, *ORMDL3*, and a slight increase in IL-17A production[40]. Furthermore, the increased expression of IL-17A has been correlated with severe asthma in humans, with increased neutrophilic infiltrates evident in mucus. These observations suggested that sphingolipid metabolism is negatively correlated with IL-17A production in asthma[40]. Since changes in the expression of *ORMDL3* could alter the amounts of all of sphingolipid downstream, increased IL-17A production may be caused by compensatory degradation of SM and Sph into glycosphingolipid. A recent study also showed that inhibition of de novo sphingolipid and glycosphingolipid biosynthesis suppresses the generation of human Th17 cells. This study also reported that sphingolipid and glycosphingolipid metabolism were enhanced in patient with type 1 diabetes (T1D)[18]. In this study, the author suggested the impact of sphingolipid pathways on CD4$^+$ T cell activation, differentiation, and effector functions in the pathogenesis of T1D. Thus, it is possible that the role of sphingolipid metabolism in Th17 cell differentiation differs in asthma and T1D[18]. Although a series of research investigated the role of sphingolipid biosynthesis pathway for Th17 and iTreg differentiation, detailed molecular mechanism has remained unclear.

In conclusion, we carried out a combined metabolomic and proteomics analysis to investigate alterations in the metabolic profiles during T cell differentiation. Our findings demonstrate that inhibition of sphingolipid or glycosphingolipid biosynthesis impairs the differentiation of Th17 and iTreg cells. Further studies focusing on the molecular mechanism behind these observations could significantly advance our understanding of the regulation of effector Th cell subsets through lipid metabolism.

## Data availability

Proteomic data used in this study has been deposited to the ProteomeXchange Consortium via the jPOST partner repository[40] (http://www.proteomexchange.org/) with the dataset identifier PXD036065. The matrix data of metabolome and lipidome analysis is provided in Supplementary Data 1. The source data for the figures is provided in Supplementary Data 1 for metabolome and lipidome analysis and Supplementary Data 2 for other data.

**Article**

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

## Acknowledgements

This work was supported by grants from the Ministry of Education, Culture, Sports, Science and Technology (MEXT Japan) (Grants-in-Aid: Grant-in-Aid for Scientific Research on Innovative Areas #18H04665, Scientific Research [B]#20H03455, Challenging Research (Exploratory) #20K21618, Early-Career Scientists #21K15476, and Young Scientists (Start-up) #21K20766). The Nakajima Foundation, TERUMO Life Science Foundation, The Tokyo Biochemical Research Foundation, Kato Memorial Bioscience Foundation, The Hamaguchi Foundation for the Advancement of Biochemistry, Suzuken Memorial Foundation, Kanae Foundation for the Promotion of Medical Science, Takeda Science Foundation, Mochida Memorial Foundation for Medical and Pharmaceutical Research, GSK Japan Research Grant 2019, SENSHIN Medical Research Foundation, Sumitomo Foundation, Koyanagi Foundation, Kishimoto Foundation 2019, Uehara Memorial Foundation,

Nakatomi Foundation, Research Foundation for Pharmaceutical Sciences Group A, Cell Science Research Foundation, The Astellas Foundation for Research on Metabolic Disorders, MSD Life Science Foundation, Public Interest Incorporated Foundation, NAGASE Science Technology Foundation, The Canon Foundation, ONO Medical Research Foundation, the Research Grant of the Princess Takamatsu Cancer Research Fund, and The Yasuda Medical Foundation, and Toray Science Foundation.

## Author contributions

T.K., R.K., M.S., Y.K., Y.H., K.I., O.O., and Y.E., conceived and directed the project, designed experiments, interpreted the results, and wrote the manuscript. T.K., R.K., M.S., A.K., Y.H., K.I., Y.K., O.O., and Y.E., designed the project, and analyzed main experiments. M.S., A.K., K.M., T.N., S.Y., S.S., A.K.H., and J.O., developed experimental protocols and performed experiments.

## Competing interests

The authors declare no competing interests.
