## [Peer review file · Communications Biology]

Reviewers' comments:

Reviewer #1 (Remarks to the Author):

In their study, Kanno et al examined the changes in metabolites and proteins in CD4 T cells differentiated into distinct subsets. They analyzed various metabolites, including sugars, TCA metabolites, amino acids, fatty acids, and lipids, and integrate it with proteomics data. Their results showed that a common feature of T cell activation and differentiation across subsets is a change in lipid composition between effector Th subsets, with the most important change being an upregulation of sphingolipid metabolism.

To further investigate the role of sphingolipid metabolism in T cell differentiation, the authors used a pharmacological inhibitor and genetic means to block this lipid biosynthetic pathway. They found that Th17 cells and iTregs cells were particularly sensitive to inhibition of this pathway and that blocking glycosphingolipid metabolism resulted in suppression of Th17 and iTreg function.

Overall, the work of Kanno et al provides insights into the role of lipid metabolism in T cell activation and differentiation. The authors' findings suggest that sphingolipid metabolism is a critical pathway for Th17 and iTreg cell development and function, and that interfering with this pathway may have therapeutic implications for autoimmune and inflammatory diseases.

Although the findings are of interest, and in line with reports showing a role for glycosphingolipid metabolism in T cell function (DOI: 10.1073/pnas.2017394118, DOI: 10.1371/journal.pone.0047650, DOI: 10.1172/JCI69571), the study might represent a valuable resource to further our understanding on the metabolic and proteomic changes of T cells upon differentiation, specially at the level of the lipidomic changes associated with T cell differentiation.

However, some specific points should be addressed at this point to further clarify the present study;

1- Metabolomics and lipidomics data normalization (related to Figure 1-3)

The study lacks clarity regarding normalization of metabolomic and lipidomic data. The methods section mentions the use of an equal number of cells across conditions (Th subsets) for metabolomic analysis. However, it is unclear how the authors normalized the lipidomic data to account for differences in cell size, which may affect lipid composition, and abundance between naive cells and the different Th cells analyzed. This issue is particularly relevant when comparing Th subsets such as in vitro generated Th1 and Th2 and Th17 and Treg cells, which have distinct size, as reported.

The same normalization problem applies to the untargeted metabolomics dataset. The authors do not describe whether they normalize the data per cell or based on proteins or other parameters. This is an important aspect to consider when analyzing the data, as differences in cell size may mask potential differences in metabolite changes between Th subsets and naive cells.

1.1 To address this issue, the authors should provide experimental clarification. For example, they could measure cell size under the different Th conditions at the time points analyzed (48h) and indicate the relevance that these changes may have in relation to the metabolites changes.

1.2 In addition, authors should discuss the impact of different cell sizes on the results or discussion section to allow for better interpretation of the data. In this way, authors can ensure that their

conclusions are not undermined by potential normalization issues related to cell size and lipid composition.

2. Sphingolipid metabolism (related to Figure 4-5)

2.1. The authors should measure the levels of sphingolipid and glycosphingolipid metabolites upon treatment of cells with myriocin or genetic knockdown of Sptlc1 or Sptlc2 to verify functional reduction of sphingolipid metabolites.

2.2 The authors should include markers of Treg cell function to better characterize the role of sphingolipid metabolism in Treg function. Authors should measure levels of the transcription factor Foxp3 and cytokine production (e.g., IL -2, IL10) when pharmacologically and genetically inhibiting sphingolipids, Figure 4h-i, 4l-m.

2.3 To investigate how sphingolipid metabolism regulates Th17 and iTreg function, it would be of interest to feed cells downstream sphingolipid metabolites (including sphingosine, ceramide, or sphingomyelins) in cells treated with myriocin. Measurement of markers of Th17 and iTreg differentiation and function under these conditions will help determine whether specific sphingolipid species play a prominent role in Th17 and Treg function.

2.4 Knockdown efficiency for Sptlc1 and Sptlc2 is not shown. The authors should show protein levels in CRISPR-Cas9 experiments.

2.5 Are sphingolipids or glycosphingolipids required for early TCR activation or later stages of T cell differentiation? Are their effects on Th17 and iTreg due to impaired TCR signaling? For example, glycosphingolipids and cholesterol may control early TCR signaling (DOI: 10.1073/pnas.2017394118). To rule out the contribution of sphingolipids or glycosphingolipids to early TCR signaling, the authors should examine markers of early TCR activation upon inhibition of sphingolipid metabolism.

2.6 Does inhibition of sphingolipid metabolism after initial T cell activation (24h, or 48h after activation) alter Th17 and iTreg function?

2.7 Pathogenic (p) and non-pathogenic (np) Th17 cells are characterized by different metabolic profiles and requirements. Do pathogenic Th17 cells rely on sphingolipid metabolism and glycosphingolipid metabolism like non-pathogenic Th17 cells?

Reviewer #2 (Remarks to the Author):

Naïve CD4⁺ T cells undergo metabolic reprogramming to meet their metabolic requirements upon exposure to antigens. However, only few studies have simultaneously evaluated the changes in protein, lipid and metabolite levels during T cell differentiation. Using a multi-omics approach, the authors demonstrate an induction of glycolysis and TCA cycle as well as an enhanced biosynthesis of amino acid and fatty acids in CD4⁺ Th cells. In response to TCR stimulation the authors found that the sphingolipid metabolism was activated in Th1, Th2, Th17, and iTreg cells, shifting the predominant sphingolipid from sphingomyelin to ceramide. However, the profiles of lipid metabolites and proteins related to lipid metabolism in Th17 and iTreg cells were significantly different from those of Th0, Th1,

and Th2 cells. Inhibition of sphingolipid and glycosphingolipid biosynthesis pathways arrested the cell cycle and cell proliferation of Th17 and iTreg cells, emphasizing the important role of the sphingolipid metabolism in T cell function.

The manuscript provides a comprehensive overview of how cellular metabolism is closely linked to the regulation of CD4⁺ T cell differentiation using very decent methods. However, there are some minor suggestions to improve the quality of the manuscript.

Figure 4:

- Does the myriocin treatment have an impact on cell viability?

The authors should include the parental live/dead gating

- Although the authors show a reduced PD1 expression on iTregs after myriocin treatment, they could perform a suppression assay to demonstrate a reduced suppressive capacity of myriocin treated iTregs

- Why does the myriocin treatment seem to have stronger effects than the genetic depletion of SPT?

The authors should include the lipid data after myriocin treatment and of sgSptlc1/sgSptlc2 cells to strengthen their data

Figure 5:

- With regard to ceramide, myriocin treatment should reduce ceramide concentrations, since it inhibits the ceramide de novo synthesis. On the other hand, Genz-123346 inhibits the glycosylceramide synthase, therefore ceramide concentrations should be elevated. Nevertheless, both inhibitors seem to have the same effect on Th17 and iTregs. Could the authors comment on possible compensatory effects?

In addition, the sphingolipid data after inhibitor treatment would be desirable to emphasize the authors' statement.

Reviewer #3 (Remarks to the Author):

Toshio Kahno et. al. investigated protein and metabolite levels during CD4 T cell differentiation of Th1, Th2, Th17 and iTreg cells. Although others have performed similar studies with RNA-sequencing technology or omics analysis, the strength of this manuscript is the simultaneous analysis of protein and metabolites. Activated T cells had a significant increase in sphingolipid biosynthesis. While this is not surprising given the requirement for membrane synthesis during rapid cell proliferation, sphingolipid biology among T cell subsets is less understood than some other metabolic pathways and is of interest to many in the immunometabolism field. The authors discovered that Th17 and iTreg cells depend on sphingolipid biosynthesis more than other T cell subsets.

The overall impression is that the dataset presented here would be useful to others if the raw data is publicly available. The conclusions about mechanistically why sphingolipid metabolism is more important for Th17 and iTreg cells than other T cells need more explanation.

1. The concentration of myriocin in these experiments is very high. 5uM and 6.25uM have been used in other papers with primary T cells (PMID 31681794, 12700647) and here the concentration is 30uM.
2. In the Results section for Figure 4, the connection from Th17/iTreg differentiation to PD-1 expression was not clearly explained and seemed random. The PD-1 data does not fit into the story of this manuscript.
3. Th17 cell metabolism has some important differences between males and females (NU Chowdhury et al, BioRxiv: <https://doi.org/10.1101/2023.02.10.527741>). Is sphingolipid biosynthesis equally required for male and female Th17 cells?
4. The formula of media and serum used in the T cell differentiation experiments needs to be included in the Methods section.
5. Th17 and iTreg cells had different ceramide species than Th0 cells although the total amount of ceramide levels was not changed. It should be tested whether specific ceramide species are responsible for their differentiation.
6. Sphingolipids and their metabolism is highly compartmentalized among the cellular organelles (PMID 27697478). It would be valuable to test whether cellular distribution of ceramides differs between Th17/iTreg and Th0 cells, potentially to help explain why they rely on sphingolipid biosynthesis more than other subsets.
7. Figure 4A. How was cell number obtained and was there consideration of cell viability?
8. Figure 4B. The x-axis should be shortened to better visualize the histograms.
9. Figure 4J. The data here is not as convincing as Figure 4F, where the cytokine signal is more robust. Does the neon transfection protocol interfere with differentiation?
10. The authors should consider the hypotheses described in the Minireview by Christopher R. Luthers et al (Frontiers in Immunology, Vol. 11, 2020) regarding biological significance of sphingolipid metabolism among T cell subsets and their findings here.
11. The discussion of TGF beta was interesting. A simple experiment would be to treat Th0, Th1, and Th2 cells with TGF beta and then test whether they become more sensitive to myriocin treatment. Furthermore, can blocking TGF beta in Th17 and iTreg cultures rescue the phenotype?

Reviewers' comments:

Reviewer #1 (Remarks to the Author):

In their study, Kanno et al examined the changes in metabolites and proteins in CD4 T cells differentiated into distinct subsets. They analyzed various metabolites, including sugars, TCA metabolites, amino acids, fatty acids, and lipids, and integrate it with proteomics data. Their results showed that a common feature of T cell activation and differentiation across subsets is a change in lipid composition between effector Th subsets, with the most important change being an upregulation of sphingolipid metabolism.

To further investigate the role of sphingolipid metabolism in T cell differentiation, the authors used a pharmacological inhibitor and genetic means to block this lipid biosynthetic pathway. They found that Th17 cells and iTregs cells were particularly sensitive to inhibition of this pathway and that blocking glycosphingolipid metabolism resulted in suppression of Th17 and iTreg function.

Overall, the work of Kanno et al provides insights into the role of lipid metabolism in T cell activation and differentiation. The authors' findings suggest that sphingolipid metabolism is a critical pathway for Th17 and iTreg cell development and function, and that interfering with this pathway may have therapeutic implications for autoimmune and inflammatory diseases.

Although the findings are of interest, and in line with reports showing a role for glycosphingolipid metabolism in T cell function(DOI: 10.1073/pnas.2017394118, DOI: 10.1371/journal.pone.0047650, DOI: 10.1172/JCI69571), the study might represent a valuable resource to further our understanding on the metabolic and proteomic changes of T cells upon differentiation, specially at the level of the lipidomic changes associated with T cell differentiation.

However, some specific points should be addressed at this point to further clarify the present study;

1- Metabolomics and lipidomics data normalization (related to Figure 1-3)

The study lacks clarity regarding normalization of metabolomic and lipidomic data. The methods section mentions the use of an equal number of cells across conditions (Th subsets) for metabolomic analysis. However, it is unclear how the authors normalized the lipidomic data to account for differences in cell size,

which may affect lipid composition, and abundance between naïve cells and the different Th cells analyzed. This issue is particularly relevant when comparing Th subsets such as in vitro generated Th1 and Th2 and Th17 and Treg cells, which have distinct size, as reported.

The same normalization problem applies to the untargeted metabolomics dataset. The authors do not describe whether they normalize the data per cell or based on proteins or other parameters. This is an important aspect to consider when analyzing the data, as differences in cell size may mask potential differences in metabolite changes between Th subsets and naïve cells.

1.1 To address this issue, the authors should provide experimental clarification. For example, they could measure cell size under the different Th conditions at the time points analyzed (48h) and indicate the relevance that these changes may have in relation to the metabolites changes.

1.2 In addition, authors should discuss the impact of different cell sizes on the results or discussion section to allow for better interpretation of the data. In this way, authors can ensure that their conclusions are not undermined by potential normalization issues related to cell size and lipid composition.

Response:

First of all, thank you for giving us constructive comments from this reviewer and the opportunity to address the concerns raised in a revised version of our manuscript.

In response to the comments, we first analyzed the cell size and the amounts of lipid droplets between naïve CD4⁺ T cells and activated CD4⁺ T cells stimulated with the α CD3/ α CD28 antibody for 48 h (Supplementary Figs. 2a and 2b, Pages 12-13). A FACS analysis showed that activated CD4⁺ T cells exhibited an almost 2-fold increase in cell size and 3-fold increase in the levels of cellular lipid droplets in comparison to the levels detected in naïve CD4⁺ T cells. Lipid droplets are composed of monoacylglycerol, diacylglycerol (DAG), triacylglycerol, and cholesteryl ester (ChE). However, when we normalized the lipidomics data based on cell size, the amounts of ChE were much lower and of DAT were unchanged in activated Th cell subsets than the levels in naïve CD4⁺ T cells (Figure for Reviewer #1). This is not consistent with our results of increased lipid droplets in activated Th cells and a previous study reporting that activated T cells increased intracellular cholesterol contents (Supplementary Fig. 2a and 2b, Pages 12-13, Reference #21). Furthermore, scaling the data

diminished the changes in the amounts of free fatty acids between activated and naïve T cells (Figure for Reviewer #1). This result is not consistent with a previous report suggesting that activated T cells exhibited augmented fatty acid uptake (Reference #10). Considering with these issues, the lipidomics data were normalized based on the cell number when comparing naïve CD4⁺ T cells and differentiated Th cell subsets. Accordingly, the metabolomics data were also normalized based on the cell number to compare naïve CD4⁺ T cells and differentiated Th cell subsets.

As Reviewer #1 pointed out, activated Th cell subsets differ in cell size; however, the difference in cell size were comparable under our experimental conditions (Supplementary Fig. 2a and 2b). Thus, to accurately compare lipid contents, the data were normalized by the signal intensity of total lipid species when comparing the lipidomics data of each Th cell subset to Th0 cells (Formula: the signal value of each lipid metabolite multiplied by 2,000,000/total signal value of lipid metabolites) (Page 16). Thus, our conclusions are not undermined by any potential normalization issues related to cell size and lipid composition.

2. Sphingolipid metabolism (related to Figure 4-5)

2.1. *The authors should measure the levels of sphingolipid and glycosphingolipid metabolites upon treatment of cells with myriocin or genetic knockdown of Sptlc1 or Sptlc2 to verify functional reduction of sphingolipid metabolites.*

Response:

As this reviewer pointed out, we checked the levels of sphingolipids and glycosphingolipids in cells treated with myriocin or lacking in the *Sptlc1/Sptlc2* genes. Lipidomics data revealed that myriocin-treated Th17 or iTreg cells had decreased levels of ceramide (Cer) and its downstream metabolites, hexosyl (Hex) Cer and DiHexCer (Figs. 4b, 4c, Supplementary Fig. 3e and 3f, Pages 19). A similar tendency was observed in *Sptlc1/Sptlc2* DKO Th17 and iTreg cells (Supplementary Figs. 5f-5i). Importantly, T cells subjected to myriocin treatment demonstrated a substantial decrease in the levels of Cer, HexCer, and DiHexCer as in comparison to T cells deficient in *Sptlc1/Sptlc2*. Our lipidomics analysis showed that the maximum reduction in Cer contents in myriocin-treated Th17 or iTreg cells was 13.5 or 4.3 log₂ FRC in comparison to control cells, respectively (Figs. 4b and 4c, Page 19). The amounts of Cer in

sg*Sptlc1/2* DKO Th17 or iTreg cells was reduced to 1.5 or 1.0 log₂ FRC of that in control cells, respectively (Supplementary Fig. 5f and 5g, Page 21). Myriocin-treated T cells also exhibited a significant decrease in the amount of glycosphingolipid, including hexosyl (Hex) Cer, and DiHexCer. The amounts of HexCer or DiHexCer in myriocin-treated Th17 or iTreg cells were reduced to 4.8 or 13 log₂ FRC of that in control cells, respectively (Supplementary Fig. 3e and 3f, Page 19). The maximum reduction of HexCer or DiHexCer contents in sg*Sptlc1/2* DKO Th17 or iTreg cells was 0.94 or 2.1 log₂ FRC in comparison to control cells, respectively (Supplementary Fig. 5h and 5i, Page 21). Taken together, T cells subjected to myriocin treatment demonstrated a substantial decrease in the levels of Cer, HexCer, and DiHexCer in comparison to T cells deficient in *Sptlc1/Sptlc2* (Supplementary Fig. 5f-5j, Page 21). This reduction likely contributed to the noteworthy decrease in IL-17A production by Th17 cells and PD-1 expression on iTreg cells. While we applied myriocin treatment from day 0, *Sptlc* gene editing was performed after 24 h TCR stimulation. These differences may be responsible for the stronger effect of myriocin on ceramide synthesis than the knockout of *Sptlc*.

2.2 *The authors should include markers of Treg cell function to better characterize the role of sphingolipid metabolism in Treg function. Authors should measure levels of the transcription factor Foxp3 and cytokine production (e.g., IL -2, IL10) when pharmacologically and genetically inhibiting sphingolipids, Figure 4h-i, 4l-m.*

Response:

As the reviewer pointed out, we examined the expression levels of FOXP3, IL-2, and IL-10 in myriocin-treated and sg*Sptlc1/2*-deficient iTreg cells.

When sphingolipid metabolism was inhibited pharmacologically or genetically, there were no significant changes in the expression of FOXP3 and IL-10 at either the mRNA or protein level (Supplementary Fig. 4b–4d and 5c-5e, Page 20). However, there was a decrease in the levels of IL-2 in myriocin-treated or sg*Sptlc1/2*-deficient iTreg cells (Supplementary Fig. 4b middle, 4c left, 4d middle, 5c middle, 5d left, and 5e middle, Page 20). We have included these results in the revised manuscript.

2.3 *To investigate how sphingolipid metabolism regulates Th17 and iTreg function, it would be of interest to feed cells downstream sphingolipid*

metabolites (including sphingosine, ceramide, or sphingomyelins) in cells treated with myriocin. Measurement of markers of Th17 and iTreg differentiation and function under these conditions will help determine whether specific sphingolipid species play a prominent role in Th17 and Treg function.

Response:

To address this, we investigated whether supplementation with sphingolipids, including sphingomyelin (SM), sphingosine (Sph), and ceramide (Cer), would restore the effect of myriocin on Th17 and iTreg cell differentiation. Although the number of myriocin-treated iTreg cells was almost unchanged by SM, Sph, Cer (d18:1/16:0), and Cer (d18:1/18:0) treatment (Supplementary Figs. 4k and 4l, Page 20), Sph and Cer (d18:1/16:0) moderately recovered the PD1 expression (Figs. 4j and 4k, Page 20). We also found that supplementation with SM, Sph, Cer (d18:1/16:0), and Cer (d18:1/18:0) moderately restored the number of myriocin-treated Th17 cells. Supplementation with these lipid species did not affect IL-17A and IL-17F production (Figs. 4l, 4m, and Supplementary Figs. 4m-4p, Pages 20-21).

These results are consistent with the observation that human differentiated Th17 and iTreg cells increased the levels of sphingolipid and glycosphingolipid and contained different composition of these lipids among CD4⁺ Th cell subsets (Reference#18, Page 29). Taken together, our data suggested that Th17 cells and iTreg cells differentially utilize sphingolipids in the regulation of cell proliferation and marker protein expression.

2.4 Knockdown efficiency for Sptlc1 and Sptlc2 is not shown. The authors should show protein levels in CRISPR-Cas9 experiments.

Response:

To address this point, we checked the knockdown efficiency of *Sptlc1* and *Sptlc2* by western blotting. sg*Sptlc1/2* Th17 or iTreg cells significantly decreased the protein levels of *Sptlc1* and *Sptlc2* (Supplementary Fig. 5a, Page 21).

2.5 Are sphingolipids or glycosphingolipids required for early TCR activation or later stages of T cell differentiation? Are their effects on Th17 and iTreg due to impaired TCR signaling? For example, glycosphingolipids and cholesterol may control early TCR signaling (DOI: 10.1073/pnas.2017394118). To rule out the contribution of sphingolipids or glycosphingolipids to early TCR signaling, the

authors should examine markers of early TCR activation upon inhibition of sphingolipid metabolism.

Response:

As this reviewer's pointed out, we analyzed whether myriocin treatment affected early TCR signaling. After 30 min TCR stimulation, inhibition of sphingolipid metabolism by myriocin in Th17 or iTreg cells did not substantially affect the phosphorylation of proteins downstream of TCR signaling (Supplementary Figs. 6l, Pages 23-24). When we inhibited glycosphingolipid metabolism by GENZ-123346, a similar tendency was observed (Supplementary Figs. 6m, Pages 23-24).

Thus, myriocin- or GENZ-123346- treatment did not substantially affect the activation status of the proteins downstream of TCR signaling.

2.6 Does inhibition of sphingolipid metabolism after initial T cell activation (24h, or 48h after activation) alter Th17 and iTreg function?

Response:

As reviewer pointed out, we analyzed the differentiation of Th17 and iTreg cells treated with myriocin for 24 or 48 hours after activation.

Although treatment with myriocin from 48 h after T cell activation failed to suppress differentiation of Th17 or iTreg cells, treatment with myriocin 24 h after TCR stimulation suppressed IL-17A production or the PD-1 expression (Supplementary Figs. 4e-4h, Page 20). However, as shown in this reviewer's comments 2.5, myriocin treatment did not substantially affect the activation status of the proteins downstream of TCR signaling.

Taken together, impaired Th17 and iTreg cell differentiation does not appear to be highly dependent on TCR stimulation.

2.7 Pathogenic (p) and non-pathogenic (np) Th17 cells are characterized by different metabolic profiles and requirements. Do pathogenic Th17 cells rely on sphingolipid metabolism and glycosphingolipid metabolism like non-pathogenic Th17 cells?

Response:

To address this reviewer's comment, we differentiated naïve CD4⁺ T cells into pathogenic (p) Th17 cells in the presence of IL-1 β , IL-6 and IL-23, and evaluated whether the inhibition of sphingolipid or glycosphingolipid metabolism suppresses the generation of pTh17 cells. Although we found that the number

of pTh17 cells decreased with the addition of myriocin or Genz-123346, the production of cytokines was not substantially affected under these conditions (Supplementary Fig. 6i-6k, Page 23). Notably, the prevention of sphingolipid metabolism did not inhibit GM-CSF production, but increased IFN γ production.

As the reviewer pointed out, pTh17 and non-pathogenic (np) Th17 cells are characterized by different metabolic profiles and requirements. Thus, different metabolic profiles in pTh17 cells may explain why the inhibition of sphingolipid or glycosphingolipid metabolism failed to suppress cytokine production, including that of GM-CSF and IFN γ . We have included these results in the revised manuscript.

Reviewer #2 (Remarks to the Author):

Naïve CD4⁺ T cells undergo metabolic reprogramming to meet their metabolic requirements upon exposure to antigens. However, only few studies have simultaneously evaluated the changes in protein, lipid and metabolite levels during T cell differentiation. Using a multi-omics approach, the authors demonstrate an induction of glycolysis and TCA cycle as well as an enhanced biosynthesis of amino acid and fatty acids in CD4⁺ Th cells. In response to TCR stimulation the authors found that the sphingolipid metabolism was activated in Th1, Th2, Th17, and iTreg cells, shifting the predominant sphingolipid from sphingomyelin to ceramide. However, the profiles of lipid metabolites and proteins related to lipid metabolism in Th17 and iTreg cells were significantly different from those of Th0, Th1, and Th2 cells. Inhibition of sphingolipid and glycosphingolipid biosynthesis pathways arrested the cell cycle and cell proliferation of Th17 and iTreg cells, emphasizing the important role of the sphingolipid metabolism in T cell function.

The manuscript provides a comprehensive overview of how cellular metabolism is closely linked to the regulation of CD4⁺ T cell differentiation using very decent methods. However, there are some minor suggestions to improve the quality of the manuscript.

Figure 4:

- Does the myriocin treatment have an impact on cell viability?

The authors should include the parental live/dead gating

- Although the authors show a reduced PD1 expression on iTregs after myriocin

treatment, they could perform a suppression assay to demonstrate a reduced suppressive capacity of myriocin treated iTregs

- Why does the myriocin treatment seem to have stronger effects than the genetic depletion of SPT?

The authors should include the lipid data after myriocin treatment and of sgSptlc1/sgSptlc2 cells to strengthen their data

Response:

First, as Reviewer #3 pointed out, since the concentration of myriocin used in our original manuscript was relatively high, we decreased the dosage of myriocin concentration from 30 μ M to 6.25 μ M, the concentration used in the previous study (Solomon, J C et al. *Cell death and differentiation*. 2003).

We first observed 6.25 μ M myriocin caused moderate decrease in the cell number (Fig. 4a, Page 19), the inhibition of cell proliferation (Supplementary Figs. 3a and 3b, Page 19), and cell cycle arrest (Figs. 4d and 4e, Page 19).

Importantly, 6.25 μ M myriocin still substantially suppressed Th17 and iTreg cell differentiation (Figs. 4f-4i, Page 20). In addition, myriocin treatment showed only slight effects on the viability of Th17 and iTreg cells (Supplementary Figs. 3c and 3d, Page 19).

We also performed a suppression assay to evaluate whether myriocin treatment affected Treg suppressive capacity. Consistent with decreased PD-1 expression and IL-2 production (Supplementary Fig. 4b middle, 4c left, 4d middle, Pages 20), myriocin-treated iTreg cells showed impaired suppressive capacity (Supplementary Figs. 4i and 4j, Pages 20). These data indicated that sphingolipid metabolism is required for a proper iTreg suppressive function.

Our lipidomics analysis showed that maximum reduction in ceramide (Cer) contents in myriocin-treated Th17 or iTreg cells was 13.5 or 4.3 log₂ FRC in comparison to control cells, respectively (Figs. 4b and 4c, Pages 19). The amounts of Cer in sgSptlc1/2 DKO Th17 or iTreg cells was reduced to 1.5 or 1.0 log₂ FRC of that in control cells (Supplementary Fig. 5f and 5g, Page 21).

Myriocin-treated T cells also exhibited a significant decrease in the amount of glycosphingolipid, including hexosyl (Hex) Cer, and DiHexCer. The amounts of HexCer or DiHexCer in myriocin-treated Th17 or iTreg cells was reduced to 4.8 or 13 log₂ FRC of that in control cells (Supplementary Fig. 3e and 3f, Page 19). The maximum reduction of HexCer or DiHexCer contents in sgSptlc1/2 DKO Th17 or iTreg cells was 0.94 or 2.1 log₂ FRC in comparison to control cells, respectively (Supplementary Fig. 5h and 5i, Page 21).

As related to this reviewer's comments to Figure 5, we consider decrease in the amounts of glycosphingolipid cause the suppression of Th17 or iTreg cells differentiation. Taken together, since myriocin-treated T cells exhibited much decrease in the levels of HexCer, and DiHexCer more than *Sptlc1/Sptlc2*-deficient T cells, there were significant reduction in IL-17A production and PD-1 expression in comparison to *Sptlc1/Sptlc2* DKO T cells. While we applied myriocin treatment from day 0, *Sptlc* gene editing was performed after 24 h of TCR stimulation. These differences may be responsible for the stronger effect of myriocin on ceramide synthesis than the knockout of *Sptlc*.

Figure 5:

- *With regard to ceramide, myriocin treatment should reduce ceramide concentrations, since it inhibits the ceramide de novo synthesis. On the other hand, Genz-123346 inhibits the glycosylceramide synthase, therefore ceramide concentrations should be elevated. Nevertheless, both inhibitors seem to have the same effect on Th17 and iTregs. Could the authors comment on possible compensatory effects?*

In addition, the sphingolipid data after inhibitor treatment would be desirable to emphasize the authors' statement.

Response:

In response to the reviewer's comments, we evaluated the amounts of sphingolipids and glycosphingolipids in Th17 or iTreg cells treated with myriocin or GENZ-123346 by lipidomics analysis.

As the reviewer pointed out, the amounts of ceramide (Cer) were substantially reduced by myriocin treatment (Figs. 4b and 4c, Page 19), while Cer accumulated following GENZ -123346 treatment (Supplementary Fig. 6e and 6f, Page 23). As shown in Fig. 5e, Th17 and iTreg cells showed higher amounts of glycosphingolipids, including hexosyl (Hex) Cer and DiHexCer, than Th1 and Th2 cells. We also found that both myriocin- and GENZ -123346-treated T cells decreased the amounts of glycosphingolipids, including HexCer and DiHexCer (Supplementary Fig. 3e, 3f, 6g and 6h, Pages 19, 23).

Thus, we consider that the decrease in the amounts of glycosphingolipids causes the suppression of Th17 or iTreg cell differentiation.

Reviewer #3 (Remarks to the Author):

Toshio Kahno et. al. investigated protein and metabolite levels during CD4 T cell differentiation of Th1, Th2, Th17 and iTreg cells. Although others have performed similar studies with RNA-sequencing technology or omics analysis, the strength of this manuscript is the simultaneous analysis of protein and metabolites. Activated T cells had a significant increase in sphingolipid biosynthesis. While this is not surprising given the requirement for membrane synthesis during rapid cell proliferation, sphingolipid biology among T cell subsets is less understood than some other metabolic pathways and is of interest to many in the immunometabolism field. The authors discovered that Th17 and iTreg cells depend on sphingolipid biosynthesis more than other T cell subsets.

The overall impression is that the dataset presented here would be useful to others if the raw data is publicly available. The conclusions about mechanistically why sphingolipid metabolism is more important for Th17 and iTreg cells than other T cells need more explanation.

1. The concentration of myriocin in these experiments is very high. 5uM and 6.25uM have been used in other papers with primary T cells (PMID 31681794, 12700647) and here the concentration is 30uM.

Response:

As this reviewer pointed out, since concentration of myriocin used in our original manuscript was relatively high, we decreased the dosage of myriocin concentration from 30 uM to 6.25 uM. We first observed 6.25 uM myriocin caused a moderate decrease in the cell number (Fig. 4a, Pages 19), the inhibition of cell proliferation (Supplementary Figs. 3a and 3b, Page 19), and cell cycle arrest (Figs. 4d and 4e, Page 19). Importantly, 6.25 uM myriocin still substantially suppressed Th17 and iTreg differentiation (Figs. 4f-4j, Page 20).

2. In the Results section for Figure 4, the connection from Th17/iTreg differentiation to PD-1 expression was not clearly explained and seemed random. The PD-1 data does not fit into the story of this manuscript.

Response:

In response to this reviewer's comment, we examined the expression levels of iTreg FOXP3, IL-2, IL-10 and PD-1 in myriocin-treated iTreg cells.

We first carefully addressed the effects of myriocin on PD-1 expression on iTreg cells. As related to this Reviewer's comment 1, since concentration of myriocin used in our original manuscript was relatively high, we decreased the dosage of myriocin concentration from 30 uM to 6.25 uM, the concentration used in the previous study (Solomon, J C et al. *Cell death and differentiation*. 2003). We found that there is a statistically significant difference in the PD-1 expression between control and myriocin-treated iTreg cells (Figs. 4h and 4i, Page 20). When sphingolipid metabolism was inhibited pharmacologically, there were no significant changes in the expression of FOXP3 and IL-10 at either the mRNA or protein level (Supplementary Fig. 4b–4d, Page 20). However, there was a decrease in the levels of IL-2 in myriocin-treated iTreg cells (Supplementary Fig. 4b middle, 4c left, 4d middle, Page 20). To investigate whether myriocin treated iTreg cells reduced suppressive capacity, we also performed a suppression assay. Consistent with decreased PD-1 expression and IL-2 production, myriocin-treated iTreg cells showed impaired suppressive capacity (Supplementary Fig. 4i and 4j, Page 20).

Taken together, we think those results are suitable for revised manuscripts and considered myriocin possibly affect PD-1 and IL-2 production in iTregs, thereby decreasing iTreg suppressive capacity.

3. *Th17 cell metabolism has some important differences between males and females (NU Chowdhury et al, BioRxiv: <https://doi.org/10.1101/2023.02.10.527741>). Is sphingolipid biosynthesis equally required for male and female Th17 cells?*

Response:

In accordance with the reviewer's comment, we evaluated whether myriocin suppressed male and female Th17 cells. Although the authors did not focus on sphingolipids or glycosphingolipids in the manuscript, our FACS analysis showed that there were no significant differences in the suppressive effects of myriocin on the production of IL-17A and IL-17F between male and female Th17 cells (Figure for Reviewer #2).

4. *The formula of media and serum used in the T cell differentiation experiments needs to be included in the Methods section.*

Response:

In response to this reviewer's comment, we included the formula of the medium and serum used in the T cell differentiation in the Methods section as follows: culture medium was RPMI-1640 medium supplemented with 55 μ M 2-mercaptoethanol, 10 μ M non-essential amino acids solution, 10 mM HEPES, 1mM sodium pyruvate, 2 mM L-glutamine, and 10 % U.S. origin FBS (Page 32).

5. *Th17 and iTreg cells had different ceramide species than Th0 cells although the total amount of ceramide levels was not changed. It should be tested whether specific ceramide species are responsible for their differentiation.*

Response:

In response to this point, we evaluated whether supplementation with ceramide (Cer) canceled the effect of myriocin on Th17 and iTreg cell differentiation. We found that Cer (d18:1/16:0) moderately recovered the expression of PD1, whereas the number of myriocin-treated iTreg cells was almost unchanged (Figs. 4k, Supplementary Figs. 4l, Page 20). In contrast, the supplementation with Cer (d18:1/16:0) and Cer(d18:1/18:0) did not restore IL-17A and IL-17F production in myriocin-treated Th17 cells. However, treatment with these lipid species moderately restored Th17 cell numbers (Figs. 4m, Supplementary Figs. 4o-4p, Pages 20-21).

Furthermore, as this Reviewer pointed out, the total amounts of ceramide (Cer) did not change in Th17 and iTreg cells in comparison to Th0 cells (Fig. 5e, Page 22). However, as shown in Fig. 5e, Th17 and iTreg cell showed higher amounts of glycosphingolipid, including hexosyl (Hex) Cer and DiHexCer, than Th1 and Th2 cells. We also found that both of myriocin- or GENZ -123346- treated T cells decreased the amounts of glycosphingolipid, including HexCer and DiHexCer (Supplementary Fig. 3e, 3f, 6g and 6h, Pages 19, 23). Thus, we consider that the decrease in the amount of glycosphingolipid causes the suppression of Th17 or iTreg cells differentiation. In addition, when we focused on the amounts of downstream metabolites of Cer, including hexosyl (Hex) Cer, DiHexCer, Th17 and iTreg cells exhibited higher amounts of HexCer (d18:1/16:0), HexCer (d18:1/18:0), and DiHexCer (d18:1/16:0) than Th0, Th1, and Th2 cells (Fig. 5g).

Taken together, these data suggested that glucosylceramide containing Cer(d18:1/16:0) and Cer(d18:1/18:0) are responsible for Th17 and iTreg differentiation.

6. *Sphingolipids and their metabolism is highly compartmentalized among the cellular organelles (PMID 27697478). It would be valuable to test whether cellular distribution of ceramides differs between Th17/iTreg and Th0 cells, potentially to help explain why they rely on sphingolipid biosynthesis more than other subsets.*

Response:

As suggested, we addressed whether the distribution of Cer in Th17 or iTreg cells differed from that in Th0 cells using NBD C6 Cer, which is a fluorescent Cer analog. Although the NBD C6 Cer fluorescent signal was mainly detected in the cellular membrane and Golgi apparatus, there were no significant changes in the distribution of NBD C6 Cer in Th17, iTreg, or Th0 cells (Figure for Reviewer #3). However, the signal intensity of NBD C6 Cer was higher in Th17 and iTreg cells than that in Th0 cells. These data suggest that Th17 and iTreg cells prefer to utilize Cer and its related downstream sphingolipids to maintain cellular homeostasis.

7. *Figure 4A. How was cell number obtained and was there consideration of cell viability?*

Response:

We diluted the cells with Trypan Blue and counted only live cells using a hemocytometer. We also found that myriocin caused only slight changes in the number of live cells (Supplementary Figs. 3c and 3d, Page 19).

8. *Figure 4B. The x-axis should be shortened to better visualize the histograms.*

Response:

As the reviewer suggested, we have modified the x-axis of Fig. 4b in the original manuscript to better visualize the histograms. We have included these data in Supplementary Fig. 3a (Supplementary Fig. 3a in the revised manuscript, Page 19).

9. *Figure 4J. The data here is not as convincing as Figure 4F, where the cytokine signal is more robust. Does the neon transfection protocol interfere with differentiation?*

Response:

To address this point, we first checked whether neon transfection slightly interfered with Th17 cell differentiation (Figure for Reviewer #4). We next

investigated the extent to which *Sptlc1/2* contributes to the production of IL-17A and IL-17F. We carefully optimized the knockout efficiency of *Sptlc1/2* and evaluated whether the deletion of these genes affects IL-17A and IL-17F production. In our optimized protocol, sg*Sptlc1/2* Th17 significantly decreased the protein levels of SPTLC1 and SPTLC2 (Supplementary Fig. 5a, Page 21). We also found that sg*Sptlc1/2* Th17 cells decreased cytokine production to almost 50% of the control Th17 cells (Fig. 4n, Page 21).

Taken together, our neon transfection protocol slightly interfered with the differentiation of Th17 cells, but we found that the genetic deletion of *Sptlc1/2* in Th17 cells reproducibly abrogated IL-17A and IL-17F production.

10. *The authors should consider the hypotheses described in the Minireview by Christopher R. Luthers et al (Frontiers in Immunology, Vol. 11, 2020) regarding biological significance of sphingolipid metabolism among T cell subsets and their findings here.*

Response:

In response to this comment, we discussed their findings regarding the biological significance of sphingolipid metabolism among T cell subsets and their findings in the revised manuscript, as described below (Reference #34, Page 30).

“As previously reported, ORMDL3 is known to negatively regulate the rate limiting enzyme of sphingolipid biosynthesis, SPT. A cohort of 17q21 risk SNP carriers showed upregulation of the locus-related gene *ORMDL3* and a slight increase in IL-17A production. Furthermore, the increased expression of IL-17A has been correlated with severe asthma in humans, with increased neutrophilic infiltrates evident in the mucus. These observations suggest that sphingolipid metabolism is negatively correlated with IL-17A production in asthma. Since changes in the expression of *ORMDL3* could alter the amounts of all of sphingolipid downstream, increased IL-17A production may be caused by compensatory degradation of SM and Sph into glycosphingolipid.”

A recent study also showed that inhibition of *de novo* sphingolipid and glycosphingolipid biosynthesis suppresses the generation of human Th17 cells (Reference#18, Pages 30-31). This study also reported that sphingolipid and glycosphingolipid metabolism were enhanced in patients with type 1 diabetes (T1D). In this study, the author suggested the impact of sphingolipid pathways on CD4⁺ T cell activation, differentiation, and effector functions in the

pathogenesis of T1D. Thus, it is possible that the role of sphingolipid metabolism in Th17 cell differentiation differs in asthma and T1D.”

11. *The discussion of TGF beta was interesting. A simple experiment would be to treat Th0, Th1, and Th2 cells with TGF beta and then test whether they become more sensitive to myriocin treatment. Furthermore, can blocking TGF beta in Th17 and iTreg cultures rescue the phenotype?*

Response:

As the reviewer pointed out, we tested whether TGF- β changes the sensitivity to myriocin. Because TGF β -treated Th0 polarization conditions are similar to iTreg polarization conditions, we investigated the effects of TGF β on Th1 or Th2 cells treated with myriocin. We first found that in the presence of TGF β , myriocin treatment caused a decrease in the number of Th1 and Th2 cells (Supplementary Fig. 3j, Page 19). However, there were no significant changes in the production of IFN γ and IL-9, which were produced by TGF β -treated Th2 cells, defined as Th9 cells (Supplementary Fig. 3k and 3l, Pages 19-20).

To investigate whether suppression of TGF β signaling in Th17 and iTreg cells restores the phenotype, we treated these Th cell subsets with SB431542, an Alk5-mediated TGF β signaling inhibitor. Treatment with SB431542 failed to restore the reduction in the number of myriocin-treated Th17 or iTreg cells (Figure for Reviewer #5). We also found that blocking TGF- β signaling strongly suppressed the production of IL-17A and IL-17F by Th17 cells (Figure for Reviewer #6). Thus, Th17 cells treated with both myriocin and SB431542 failed to produce these cytokines (Figure for Reviewer #6). Although SB431542 clearly suppressed the phosphorylation of Smad2/Smad3, PD-1 expression was not significantly altered in iTreg cells (Figures for Reviewer #7 and #8). Importantly, the impairment of the PD-1 expression in myriocin-treated iTreg cells was not recovered by treatment with SB431542 (Figure for Reviewer #7).

In the original manuscript, we considered the possibility that Smad2/Smad3 contributes to the formation of unique sphingolipid and glycosphingolipid profiles in Th17 and iTreg cells. However, the PD-1 expression was not restored by inhibition of Smad2/Smad3 signaling. It has also been reported that aberrant TGF β -mediated Alk5/Smad2 or Alk1/Smad1 pathways contribute to the disruption of sphingolipid metabolism during intrauterine growth restriction (IUGR) (Reference #30). Alk5/Smad2 contributes

to Cer breakdown, resulting in the accumulation of downstream glycosphingolipids (Reference#30). In addition, impaired TGF β signaling *via* Alk1/Smad1 contributes to sphingosine buildup in IUGR (Reference #30, Pages 28-29). Thus, we considered the possibility that the Alk1/Smad1 signaling pathway, in addition to Smad2/3, contributes to the construction of unique sphingolipid and glycosphingolipid profiles in Th17 and iTreg cells. Indeed, SB431542 did not affect the phosphorylation levels of Smad1 and Smad8 (Figure for Reviewer #8). We discussed these issues in the revised manuscript (Pages 28-29).

Figures for Reviewer

#1

Without data scaling

With data scaling

#1, Lipidomics data with and without scaling

Lipidomics data shows the amounts of ChE, DAG and FA in naïve CD4 T cells, Th0, Th1, Th2, Th17, and iTreg cells with or without data scaling (Without data scaling : A-C, With data scaling : D-F)

#2

#2, Analysis of effects of myriocin on sex differences in Th17 cells differentiation.

FACS analysis investigate whether effects of myriocin on IL-17A and IL-17F production was differ in male and female mouse (A). Summarized data was shown (B).

#3

#3, Analysis of ceramide distribution in Th17, iTreg and Th0 cells.

Immunofluorescence microscopy and flowcytometry show the distribution (A) or fluorescence intensity (B) of NBD C6 Cer in Th0, Th17, and iTreg cells.

#4

#4, Analysis of NEON electroporation on Th17 cell differentiation.

FACS analysis shows IL-17A and IL-17F production in Th17 cell performed with or without NEON electroporation (A). Summarized data was shown (B).

#5

#5, Investigation of effects TGF β signaling inhibition on Th17 and iTreg cell proliferation in the presence of myriocin.

Cell number was calculated after 3 days culture of Th17 and iTreg polarization condition in the presence of myriocin and TGF β R1 inhibitor (A).

#6

#6, Investigation of effects TGF β signaling inhibition on Th17 cell differentiation in the presence of myriocin.

FACS analysis shows the IL-17A and IL-17F production in the presence of myriocin and TGF β R1 inhibitor (A). Summarized data was shown (B).

#7

#7, Investigation of effects of TGF β signaling inhibition on iTreg cell differentiation in the presence of myriocin.

FACS analysis shows the PD-1 expression in the presence of myriocin and TGF β R1 inhibitor (A). Summarized data was shown (B).

#8

#8, Investigation of phosphorylation of Smad2/3 or Smad1/8 on iTreg cell differentiation in the presence of myriocin with or without TGFβR1 inhibitor.

FACS analysis shows the phosphorylation levels of pSmad2/3 (A) or pSmad1/8 on iTreg cells in the presence of myriocin and TGFβR1 inhibitor. Summarized data was shown in right side of each figure.

REVIEWERS' COMMENTS:

Reviewer #1 (Remarks to the Author):

This reviewer appreciates the efforts from the authors in terms of experiments and discussion to address my previous concerns. This reviewer considers that the manuscript is greatly improved and there are not any other major concerns.

I believe that this study will be a very valuable resource for many researchers interested in studying the metabolic basis of T cell differentiation and, in particular, understanding how lipid metabolism is regulated in Th subsets and controls Th specification.

Reviewer #2 (Remarks to the Author):

After carefully reviewing the initial submission and the revisions made by the authors, I am pleased to report that the manuscript has been substantially improved and is now suitable for publication.

The authors have addressed the concerns raised during the initial review process, effectively enhancing the quality of the manuscript. This work will represent a good contribution to our understanding how sphingolipids and cellular metabolism modulate T cell function and differentiation.

Reviewer #3 (Remarks to the Author):

The revisions to the manuscript have sufficiently addressed all concerns and increased the quality of the paper significantly. Well done.